# Lipid transporters E-Syt3 and ORP5 regulate epithelial ion transport by controlling phosphatidylserine enrichment at ER/PM junctions

Paramita Sarkar[1,6], Benjamin P Lüscher[1,4,6], Zengyou Ye [1,5,6], Woo Young Chung[1], Ava Movahed Abtahi [1], Changyu Zheng[1], Min Goo Lee [2], Árpád Varga [3], Petra Pallagi[3], József Maléth [3], Malini Ahuja[1] & Shmuel Muallem [1✉]

## Abstract

Endoplasmic reticulum/plasma membrane (ER/PM) junctions are a major site of cellular signal transduction including in epithelia; however, whether their lipid membrane environment affects junctional ion transporters function remains unclear. Here, we show that epithelial secretion is governed by phosphatidylserine (PtdSer) levels in ER/PM nanodomains, specified by the antagonistic action of the lipid transfer proteins E-Syt3 and ORP5, which transduce cAMP signals to the chloride channel CFTR and activate the sodium-bicarbonate cotransporter NBCe1-B by IRBIT. Lipid transfer by E-Syt3, along with restricted plasma membrane localization by the E-Syt3 C2C domain, are essential for E-Syt3 function, as removal of PtdSer from junctions by E-Syt3 dissociated the cAMP signaling pathway complex, preventing CFTR activation, and prevented NBCe1-B activation by IRBIT. CFTR and NBCe1-B PtdSer sensor domains responded to PtdSer reduction by E-Syt3; which was reversed by exogenous PtdSer or by PtdSer supplied by ORP5. In mice, E-Syt3 depletion improved chloride flux and fluid secretion in salivary glands and isolated pancreatic ducts. These findings provide a framework for understanding the role of junctional lipids in the assembly of functional ion protein complexes and cellular communication at epithelial signaling hubs.

**Keywords** E-Syt3/ORP5; Junctional Phosphatidylserine; CFTR/NBCe1-B; Regulation
**Subject Categories** Digestive System; Membranes & Trafficking

## Introduction

Vectorial fluid secretion constitutes a fundamental physiological function that requires the formation of transporters complexes and signal transduction at specialized plasma membrane subdomains (Lee et al, 2012). In recent years, it has become clear that the ER/PM junctions form sites for the organization of protein and signaling complexes at which lipids and ions are exchanged between cellular membranes and compartments (Arora et al, 2022; Dickson, 2022; Machaca, 2020; Muallem et al, 2017; Prinz et al, 2020). The ER/PM junctions are dynamic structures assembled by tether proteins that are often lipid transfer proteins (LTP), such as the three extended synaptotagmins (E-Syts) (Giordano et al, 2013; Saheki and De Camilli, 2017) and the oxysterol-binding protein related proteins ORP5 and ORP8 (Chung et al, 2015). All E-Syts have an ER-anchored N-terminus transmembrane domain that is followed by lipid transfer SMP (synaptotagmin-like mitochondrial lipid-binding protein) domain and 5 (E-Syt1) or 3 (E-Syt2 and E-Syt3) $Ca^{2+}$ and polar lipid-binding C2 domains (Giordano et al, 2013; Saheki and De Camilli, 2017). The ORPs have an N-terminus PH domain (PHD) and associated polybasic sequence, a lipid transfer OSBP-related domain (ORD) and a C-terminus transmembrane domain located in the ER. In vitro studies using reconstituted isolated SMP domains and purified cytoplasmic E-Syt1 into vesicles of various lipid compositions reported similar transfer of all glycerophospholipids, including phosphatidylcholine (PtdC), phosphatidylethanolamine (PtdEtn), phosphatidylserine (PtdSer), $PI(4,5)P_2$, and cholesterol with minimal selectivity (AhYoung et al, 2015; Bian et al, 2018; Yu et al, 2016). In vivo measurements showed that E-Syt1+E-Syt2 affects the recovery of plasma membrane (PM) diacylglycerol (DAG) and externalize PtdSer (Bian et al, 2018; Sassano et al, 2023; Thomas et al, 2022). The yeast homologs of the E-Syts, the tricalbins regulate PM PtdSer homeostasis (Thomas et al, 2022). The ORPs function as PI(4)P/PtdSer exchangers

[1]Epithelial Signaling and Transport Section, National Institute of Dental Craniofacial Research, National Institutes of Health, Bethesda, MD 20892, USA. [2]Department of Pharmacology, Brain Korea 21 PLUS Project for Medical Sciences, Severance Biomedical Science Institute, Yonsei University College of Medicine, Seoul 03722, Korea. [3]First Department of Internal Medicine, University of Szeged, Szeged, Hungary. [4]Present address: Department of Hematology and Central Hematological Laboratory, Inselspital, Bern University Hospital, University of Bern, Bern, Switzerland. [5]Present address: Neurocircuitry of Motivation Section, Behavioral Neuroscience Research Branch, Intramural Research Program, National Institute on Drug Abuse, National Institutes of Health, Baltimore, MD 21224, USA. [6]These authors contributed equally: Paramita Sarkar, Benjamin P Lüscher, Zengyou Ye. ✉E-mail: shmuel.muallem@nih.gov

(Chung et al, 2015; Sohn et al, 2018), but with opposite directionality. ORP8 transfers PtdSer to the ER and PI(4)P to the plasma membrane (PM), while ORP5 transfers PI(4)P to the ER and PtdSer to the PM (Chung et al, 2023).

The specific impact of E-Syts on the lipid composition of ER/PM junctions, and their role in forming and transmitting signals that regulate ion transporters and epithelial secretion, remain unclear. To address these questions, we focused on the transporters mediating epithelial fluid and $HCO_3^-$ secretion, which are essential for sustaining life. Aberrant $HCO_3^-$ secretion is associated with virtually all epithelial diseases, including cystic fibrosis (CF) (Angyal et al, 2021; Choi et al, 2001), pancreatitis (Angyal et al, 2021; Zeng et al, 2017), inflammatory bowel disease (Shin et al, 2020) and anomalous fertilization (Liu et al, 2012). Epithelial $HCO_3^-$ secretion is a vectorial process that entails $HCO_3^-$ entry across the basolateral membrane and $HCO_3^-$ exit across the luminal membrane (Lee et al, 2012). $HCO_3^-$ enters epithelial cells by the basolateral electrogenic $Na^+/2HCO_3^-$ cotransporter NBCe1-B and exits by the luminal $Cl^-$ and $HCO_3^-$ channel Cystic Fibrosis Transmembrane Conductance Regulator (CFTR) and the electrogenic $Cl^-/2HCO_3^-$ exchanger SLC26A6 (Ko et al, 2004; Lee et al, 2012; Wang et al, 2006) (see model in Fig. 9F below). The energy-demanding fluid and $HCO_3^-$ secretion is a highly regulated process that is quiescent at the resting state and leaked $HCO_3^-$ is salvaged to maintain an acidic luminal environment, as observed with the pancreatic juice (Lee et al, 2000) and saliva (Luo et al, 2001). The secretory state is triggered by a combination of sympathetic and parasympathetic stimuli to activate the duct cells NBCe1-B and CFTR (Hong et al, 2014; Lee et al, 2012). NBCe1-B has N-terminus autoinhibitory domain (AID) and is activated by IRBIT ($IP_3$ receptors binding protein released with $IP_3$). IRBIT interacts with the AID to dissociate it and set the NBCe1-B active conformation (Hong et al, 2013; Shirakabe et al, 2006; Yang et al, 2009). CFTR is activated by the adenylyl cyclase (AC)/AKAP/cAMP/PKA pathway (Kunzelmann and Mehta, 2013; Omar and Scott, 2020). Activation of NBCe1-B and CFTR requires the formation of several signaling complexes at the juxtaposed basolateral and luminal membranes. The role of localization at the ER/PM junctions for the formation of the complexes and activation of the proteins is not known and is addressed in the present studies.

Lipids play a crucial role in regulating the activity of numerous channels and transporters (Levental and Lyman, 2023; Schmidpeter et al, 2022; Thompson and Baenziger, 2020). Increasing evidence suggests that defined PM lipid domains have important roles in regulating cellular signaling and ion channel activity. For example, $PI(4,5)P_2$ metabolism appears to be compartmentalized in PM microdomains (Kefauver et al, 2024; Myeong et al, 2021), which is required for the regulation of STIM1-Orai1 function (Maleth et al, 2014), TRP channels (Bobkov and Semenova, 2022; Cai and Chen, 2023; Liu et al, 2022) and the function of both CFTR (Himmel and Nagel, 2004) and NBCe1-B (Hong et al, 2013; Thornell et al, 2012). The role of other lipids, particularly the negatively charged PtdSer, in the assembly of signaling complexes and regulation of transporters activity are not known. Our findings show that only PtdSer is indispensable for the assembly of the regulatory complexes formed by CFTR and NBCe1-B to activate them. Examining the role of lipid transfer proteins shows that the E-Syt1 increases while E-Syt2 and E-Syt3 reduce PM PtdSer. However, only a reduction in PtdSer by E-Syt3 reduced the activity of CFTR

and NBCe1-B. The inhibition by E-Syt3 requires both lipid transfer by the SMP domain and binding of the third E-Syt3 C2 domain (E3C2C) to PM $PI(4,5)P_2$. Remarkably, the E3C2C domain was sufficient to target the E-Syt1, E-Syt2, and E-Syt3 SMPs to alter the lipid composition of the transporter's domain at the ER/PM junctions and mediate their regulation, like the intact E-Syt3. The specific E-Syt3 and E3C2C + SMP reduction of PM PtdSer was reversed by supplementing the PM with exogenous PtdSer, but not with PtdE, PtdC or $PI(4,5)P_2$. Moreover, increased junctional PtdSer by ORP5 reversed inhibition by E-Syt3. In addition, acutely targeted SMP and the specific PtdSer hydrolyzing mouse PLA1a1 to the PM with the FRB/FKBP/rapamycin system reproduced the effect of E-Syt3. CFTR sensed PM PtdSer by its lasso domain while NBCe1-B sensed PtdSer by its AID. PtdSer regulated CFTR by mediating the assembly of the CFTR-AC-AKAP5-PKAc complexes and activation of CFTR by the cAMP/PKA pathway. PtdSer regulated NBCe1-B by mediating the formation of the NBCe1-B-IRBIT complexes. The physiological significance of the regulation by E-Syt3/PtdSer is established in mice by demonstrating inhibition and enhanced salivary $Cl^-$ and $Na^+$ absorption by transgenic overexpression and silencing of E-Syt3, respectively, in salivary gland ducts and augmented fluid secretion by depletion of E-Syt3 in the microdissected sealed pancreatic duct. Our findings reveal novel mechanisms for the regulation of ion transporters at the ER/PM junctions by PtdSer and reveal an unexpected role for PtdSer nanodomain at the PM formed by E-Syt3/ORP5 to determine the function of epithelial fluid and $HCO_3^-$ secretion. These findings have major implications for diseases associated with the function of CFTR and NBCe1-B.

## Results

### E-Syt3, but not E-Syt1 and E-Syt2, inhibits CFTR and NBCe1-B activity

The importance of lipids in membrane contacts sites in the function of ion transporters have not been explored before. To address this issue, we focused on epithelial fluid and $HCO_3^-$ secretion that entails NBCe1-B-mediated basolateral $HCO_3^-$ influx and CFTR-dependent luminal $HCO_3^-$ exit (Lee et al, 2012). Prominent tethers at ER/PM contact sites are the lipid transfer proteins (LTP) extended synaptotagmins (E-Syts) (Giordano et al, 2013; Saheki and De Camilli, 2017). The domain boundaries and predicted structure of E-Syt3 are shown in Fig. 1A. Measurement of CFTR current showed that when it is co-expressed with E-Syt3, but not with E-Syt1 or with E-Syt2, at a CFTR/E-Syts ratio of 1, the current is prominently inhibited (Fig. 1Bi–iii). Moreover, depletion of the native E-Syt3 by siRNA increased the CFTR current (Fig. 1C). Measurement of the NBCe1-B current at NBCe1-B/E-Syts ratio of 1 showed similar prominent inhibition by E-Syt3, while E-Syt1 and E-Syt2 had no effect (Fig. 1Di–iii) and depletion of E-Syt3 increased the NBCe1-B activity (Fig. 1E). However, when E-Syt2 was expressed at a high E-Syt2/CFTR ratio of 3:1 it partially activated, rather than inhibited, CFTR (Appendix Fig. S1A) and expression at a 1CFTR/1E-Syt3/3E-Syt2 ratio showed that E-Syt3 dominated over E-Syt2 effect (Appendix Fig. S1B). E-Syt2 had no effect on NBCe1-B activity at NBCe1-B/3E-Syt2 ratio in the presence or absence of E-Syt3 (Appendix Fig. S1C). Therefore, in

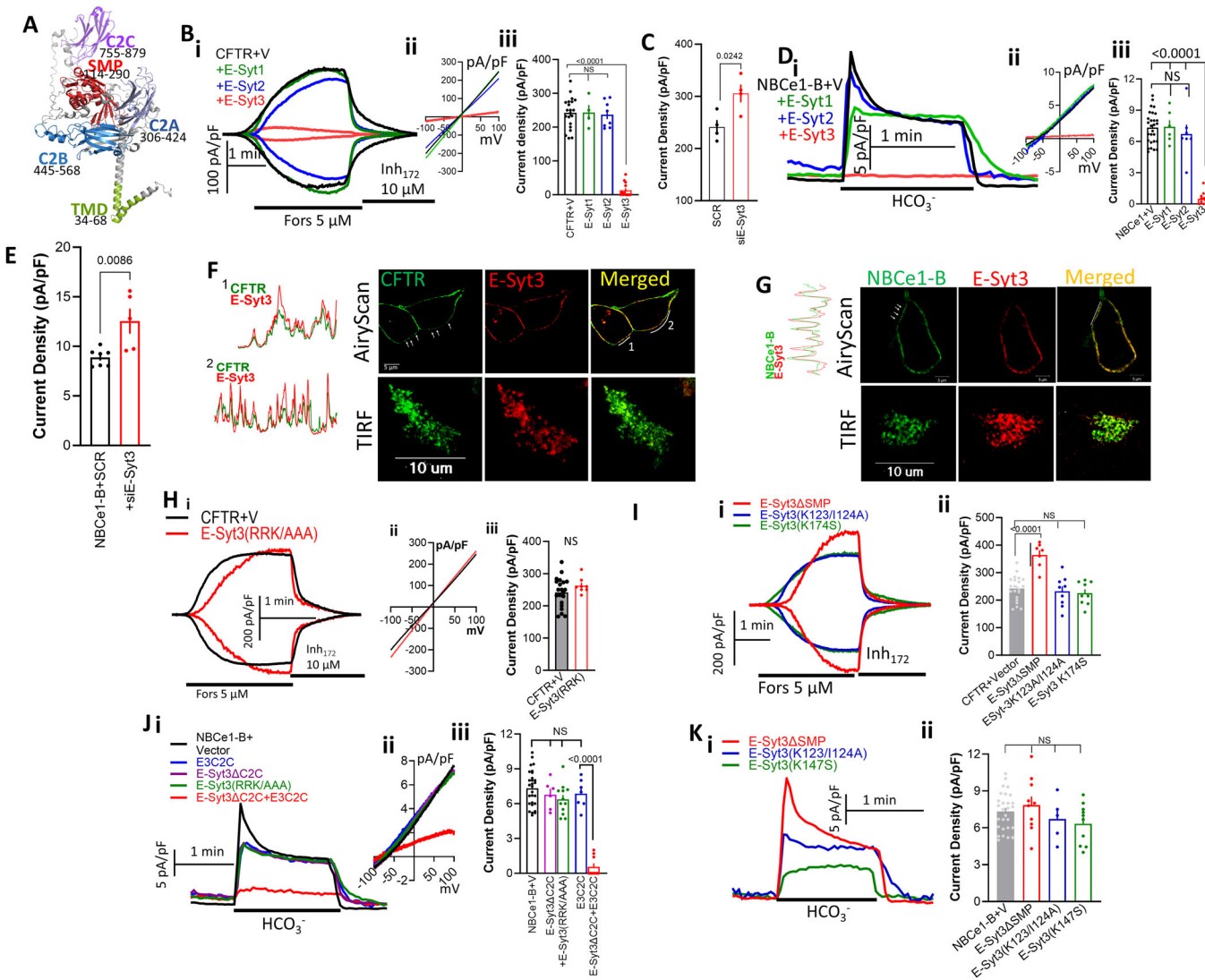

**Figure 1. E-Syt3, but not E-Syt1 and E-Syt2, inhibits CFTR and NBCe1-B activity that requires E3C2C and lipid transfer by the SMP domain.**

CFTR current was measured in response to stimulation with 5 μM forskolin and 100 μM IBMX (F/I) while NBCe1-B activity was initiated by switching the cells from a solution buffered with Hepes to a solution containing 25 mM $HCO_3^-$. (A) A model of E-Syt3 structure and domains as predicted by Robetta. (Bi–iii) CFTR current in HEK293T cells expressing CFTR and vector (black), E-Syt1 (green), E-Syt2 (blue) or E-Syt3 (red). Here and in other Figures (i) time course, (ii) example I/V, (iii) current density. The vector results (mean ± s.e.m) are from 9 transfections and 21 cells, and the E-Syt1 2/4, E-Syt2 3/8, and E-Syt3 4/13 transfections/cells. The number of Transfections and Cells are listed below as T/C. (C) Cells treated with scrabbled (SCR) or E-Syt3 siRNA were used to measure CFTR current in 2/4 transfections/cells and are shown as mean ± s.e.m. (Di–iii) NBCe1-B current in HEK293T cells expressing NBCe1-B and vector (black 10/28 T/C), E-Syt1 (green, 3/6 T/C), E-Syt2 (blue, 3/7 T/C) or E-Syt3 (red, 5/12 T/C) and are shown as mean ± s.e.m. (E) Cells treated with scrambled (SCR, 2/7 T/C) or E-Syt3 siRNA (2/5 T/C) were used to measure NBCe1-B current and are shown as mean ± s.e.m. (F, G) Example images of HEK293T cells expressing CFTR and E-Syt3 (F) or NBCe1-B and E-Syt3 (G), as indicated in the images. The upper images are superresolution AiryScan images. The punctate patterns are marked by arrows in the green images and the scanned regions by white lines. The lower images are TIRF microscopy images of expression at the ER/PM junctions. (Hi–iii) Current in cells expressing CFTR+Vector (black, 9/21 T/C) or E-Syt3(R852A/R853A/K854A) (red, 2/7 T/C) and are shown as mean ± s.e.m. (Ii–ii) Current in cells expressing CFTR + E-Syt3(ΔSMP) (red, 2/6 T/C), E-Syt3(K123A/I124A) (blue 3/9T/C), or E-Syt3(K147S) (green 3/9 T/C) and are shown as mean ± s.e.m. (Ji–iii) NBCe1-B current in cells expressing NBCe1-B (black, 10/28 T/C) and E2C2C (blue 2/7 T/C), E-Syt3(ΔC2C) (magenta 2/6 T/C), E-Syt3(R852A/R853A/K854A) (green 3/11 T/C), or E-Syt3(ΔC2C) and E3C2C (red 3/9 T/C) and are shown as mean ± s.e.m. (Ki–ii) NBCe1-B current in cells expressing NBCe1-B and E-Syt3(ΔSMP) (red 3/10 T/C), E-Syt3(K123A/I124A) (blue 2/5 T/C), or E-Syt3(K147S) (green 3/10 T/C) and are shown as mean ± s.e.m. All P values were determined by unpaired Student t test. Source data are available online for this figure.

the present studies, we focused on the regulation of the transporters by E-Syt3. The selective effect of E-Syt3 is not due to a different total protein expression of E-Syt3 compared with the other E-Syts (Appendix Fig. S1D), different localization (Appendix Fig. S1E) or due to differential interaction of the E-Syts with the transporters since interaction of E-Syt2 and E-Syt3 with CFTR and NBCe1-B

was similar whether measured by FRET (Appendix Fig. S1F) or by Co-IP (Appendix Fig. S1G). Finally, Fig. 1F,G shows that the nearly complete inhibition of the transporters by E-Syt3 was not due to inhibition of their expression at the plasma membrane (PM), as determined by superresolution AiryScan microscopy (upper images). Assay of junctional expression in live cells by TIRF

microscopy further supports no effect of either E-Syt3 expression (lower images in Fig. 1F,G and summary in Appendix Fig. S1H,I) or depletion of the level of CFTR and NBCe1-B in the TIRF field (Appendix Fig. S1H,I).

## E-Syt3 function requires lipid binding by the SMP domain and localization by the C2C domain

To determine the structural components of E-Syt3 required for the inhibition, we mutated residues suggested by the structure of E-Syt2 SMP domain to participate in SMP dimerization and lipid binding (Schauder et al, 2014). The alignment in Appendix Fig. S2A identified the equivalent K123/I124 and K147 in the hydrophobic pocket of the E-Syt3 SMP domain. C2 domains are $Ca^{2+}$ and phospholipids binding modules, where phospholipid binding is mediated by positively charged residues (Corbalan-Garcia and Gomez-Fernandez, 2014; Corbalan-Garcia et al, 2007). The E-Syt3 C2C domain (E3C2C) is its $PI(4,5)P_2$-binding module (Giordano et al, 2013) mediated by R852/R853/K854 (Appendix Fig. S2A). Mutation of the SMP and E3C2C binding sites or deletion of the entire E-Syt3 SMP domain (E-Syt3ΔSMP) did not prevent localization of E-Syt3 at the ER/PM junctions, while mutating the E3C2C domain (E-Syt3(RRK/AAA)) resulted in ER and cytoplasmic localization (Appendix Fig. S2B–E). The E-Syt3 and CFTR co-immunoprecipitation (Co-IP) was not affected by the SMP K123A/I124A mutant but was nearly eliminated by the E-Syt3(RRK/AAA) mutant (Appendix Fig. S3A). Moreover, the Co-IP of E3C2C domain with CFTR (Appendix Fig. S3B), suggests that E3C2C mediates E-Syt3 co-localization with CFTR. Indeed, the E-Syt3(RRK/AAA) (Fig. 1Hi–iii) and SMP (Fig. 1Ii–ii) mutants did not inhibit CFTR current. Similar findings were made with NBCe1-B that interacted with E-Syt3 and E-Syt2 (Appendix Fig. S3C), and E3C2C (Appendix Fig. S3D), while E-Syt3(RRK/AAA) and E-Syt3ΔC2C (Fig. 1Ji–iii) and SMP mutants (Fig. 1Ki–ii) failed to inhibit the NBCe1-B current. Interestingly, co-expression of E-Syt3ΔC2C + E3C2C as two separate proteins restored inhibition of the NBCe1-B current (Fig. 1Ji–iii, red traces, and columns).

To further demonstrate the role of the SMP domain we switched between the E-Syt2 and E-Syt3 SMP domains (Appendix Fig. S3E). Appendix Fig. S3E shows that E-Syt3 with E-Syt2 SMP domain (E-Syt3(E2SMP)) expressed at the cell periphery, as does E-Syt3 (Fig. 1F,G). Appendix Fig. S3Fi–ii shows that inhibition of CFTR track with E3C2C rather than with the SMPs, indicating the E-Syt2 SMP is functional and can mediate the inhibition of the transporters when targeted to the domain at which E3C2C binds to the PM. To further analyze the functionality of the E-Syts SMP and C2C domains, we expressed the ER-anchored SMP domains of the E-Syts (named E(1–3)SMPX constructs) and the cytoplasmic SMP domains (named SMP1-3) alone and together with E-Syt2 or E-Syt3 C2C domains (named E2C2C and E3C2C, Fig. 2A). Both E2C2C and E3C2C localized to the cell periphery (Appendix Fig. S4A,B) and E3C2C recruited part of the cytoplasmic E3SMP domain to the PM (Appendix Fig. S4Ci–ii). Moreover, the E3C2C domain enhanced the interaction of the E3SMP with CFTR (Appendix Fig. S4D,F) and NBCe1-B (Appendix Fig. S4E,F). Both CFTR and NBCe1-B have C-terminal PDZ ligand that interacts with PDZ domains present in scaffolding proteins to facilitate their interaction with other proteins (Ahn et al, 2001; Ko et al, 2004; Weinman et al, 2001). However, interaction with PDZ scaffolds was

not essential for the inhibition of CFTR and NBCe1-B by E-Syt3 (Appendix Fig. S4G,H).

The functional consequences of the interaction of the E(1–3) SMPX and SMP1-3 with E3C2C are shown in Fig. 2B–E. The E(1–3)SMPX alone had no effect on transporters activity. However, all the ER-anchored SMPX domains similarly inhibited CFTR (Fig. 2Bi–iii) and NBCe1-B (Fig. 2Ci–ii) when co-transfected with E3C2C. This was specific to E3C2C since expression of E2C2C with E3SMPX did not inhibit the transporters. We note that the C2C domains of E-Syt2 and E-Syt3 target their SMP domains to the plasma membrane. However, despite the C2C domains high homology, they are not identical and appear to target E-Syt2 and E-Syt3 to different PM subdomains (Giordano et al, 2013). In fact, the expression of E2C2C with E3SMPX significantly increased the activity of CFTR and NBCe1-B (red traces and columns in Fig. 2B,C), possibly by reversing inhibition by the native E-Syt3. Hence, the SMP domains of all E-Syts are functional, but to affect the activity of the transporters, they must be targeted at a precise subdomain where E3C2C interacts with the PM. Finally, the crystal structure of SMP domains showed that it dimerizes to form a hydrophobic lipid transfer tunnel (Jeong et al, 2017; Schauder et al, 2014). Therefore, we tested whether the isolated E3SMP alone can function in vivo and how its activity is affected by E3C2C. When expressed at high levels, the E3SMP was able to partially inhibit CFTR (Fig. 2Di–iii) and NBCe1-B (Fig. 2Ei–iii) activity. Significantly, the effectiveness of E3SMP was prominently increased by co-expression with E3C2C and it was reversed by E2C2C, further demonstrating the importance of E3C2C junctional subdomain and the importance of the privileged localization of the C2 domains. Although the SMP domain is not anchored in the ER it interacts with the ER and PM by forming dimers that have positively charged regions at their apex that mediate binding to negatively charged membrane phospholipids like PtdSer and $PI(4,5)P_2$ to allow lipid transport (Wang et al, 2023). Indeed, the SMP(K207E/I208E) mutant that inhibited lipid transfer by the purified SMP domain (Wang et al, 2023), when expressed with E3C2C failed to inhibit the CFTR current (Appendix Fig. S10C).

## E-Syt3 regulates the transporters by affecting Junctional PtdSer

Measurement of lipid transport in artificial vesicle assays showed that the SMP domain equally transports all glycerophospholipids, PtdSer, PtdE, PtdC, $PI(4,5)P_2$ and diacylglycerol (DAG) (AhYoung et al, 2015; Bian et al, 2018; Yu et al, 2016). In intact cells, E-Syt1 facilitated the transfer of DAG from the PM to the ER, and cells lacking all E-Syts showed impaired externalization of PM PtdSer (Bian et al, 2018; Saheki et al, 2016). Therefore, we used biosensors (Appendix Fig. S5A–C) to investigate the effect of the E-Syts on PM PtdSer and other phospholipids as a potential mechanism underlining the function of E-Syt3. We verified that the probes detect the lipids in the ER/PM junctions by determining good co-localization with MAPPER (Appendix Fig. S5D,E), a junctional marker (Chang et al, 2013). First, we measured the effect of the E-Syts on junctional PtdSer, PI(4)P and $PI(4,5)P_2$ and PtdSer externalization. Figure 3A–C shows that E-Syt1 slightly increased, while E-Syt2 and E-Syt3 exhibited comparable reduction in junctional PtdSer, PI(4)P and $PI(4,5)P_2$. The effects of E-Syt3 were reversed by depletion of the native E-Syt3 and were prevented by the E-Syt3

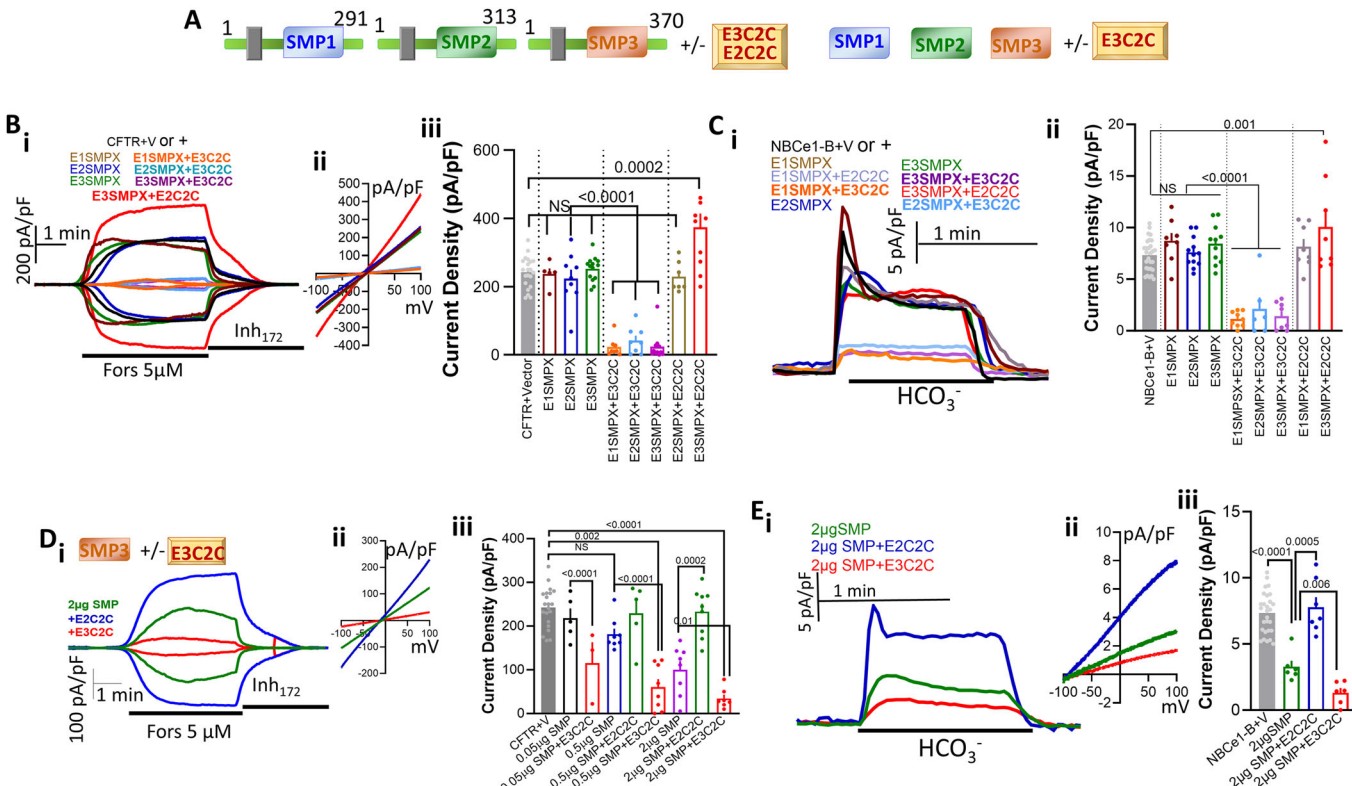

**Figure 2. E-Syt3 E3C2C targets E-Syt1 and E-Syt2 SMPs to a domain required to inhibit CFTR and NBCe1-B activity.**

(A) The boundaries of E-Syt1X, E-Syt2X and E-Syt3X and the SMP and E3C2C combinations tested. (Bi–iii, Ci–ii) CFTR (B) and NBCe1-B (C) currents in HEK293T cells expressing Vector (black), E1SMPX (purple), E2SMPX (blue), E3SMPX (green), SMP1X + E3C2C (orange), E2SMPX + E3C2C (turquoise) or E3SMPX + E3C2C (magenta). Results for CFTR are from 2/5 T/C for E1SMPX and at 3-4/8-14 T/C for the other conditions. Results for NBCe1-B are from 2/6 T/C for E2SMPX + E3C2C and at 3-4/8-12 T/C for the other conditions and are shown as mean ± s.e.m. (Di–iii, Ei–iii) CFTR (D) and NBCe1-B (E) currents in HEK293T cells expressing 2 μg SMP (turquoise) together with E2C2C (green) or E3C2C (red). Results for CFTR are from 2/3-5 T/C for 0.05 μg SMPX + E3C2C and 0.5 μg SMP + E2C2C and at 2-3/8-9 T/C for the other conditions. Results for NBCe1-B are from 2/6-7 T/C and are shown as mean ± s.e.m. All *P* values were determined by unpaired Student *t* test. Source data are available online for this figure.

SMP and E3C2C mutants (Fig. 3D,E), indicating that the native E-Syt3 actively modulates junctional lipids. E-Syt1 is activated by increasing cytoplasmic $Ca^{2+}$. However, even when activated by CPA treatment, E-Syt1 failed to affect CFTR activity (Appendix Fig. S6). E-Syt1 in the presence of E-Syt2 causes a small PtdSer externalization (Bian et al, 2018). We did not observe significant PtdSer externalization by expression of any of the E-Syts, or their effect on the massive externalization caused by treating the cells with CPA (Appendix Fig. S7A–C). Therefore, the prominent reduction in junctional PtdSer caused by E-Syt2 and E-Syt3 is by a mechanism different from PtdSer externalization. The mechanism(s) by which the E-Syts control junctional lipid composition remain to be defined.

Since E-Syt3 prominently changes junctional PtdSer and PI(4,5)$P_2$ and PI(4,5)$P_2$ has been shown to regulate many channels, we tested whether acutely supplementing the cells with PtdSer, PI(4,5)$P_2$ and other glycerophospholipids through the patch pipette can reverse the inhibition by E-Syt3. In preliminary experiments, we tested different concentrations of PtdSer and infusion time on the integrity of the cells, the whole cell configuration, (maintained membrane resistance and capacitance) and current restoration. We used brain lipids to ensure the presence of all native lipid isoforms

and with the in vivo ratio. The optimal conditions found are using 10 μg/ml lipids and infusion for up to 8 min prior to stimulation with forskolin (CFTR) or the addition of $HCO_3^-$ (NBCe1-B). Figure 3Fi–iii shows that infusing HEK293T cells expressing only CFTR with 10 μg/ml of the major PM lipids PtdSer, PtdC or PtdE had no effect on basal or forskolin-activated current. Most notably, infusing the cells with PtdSer completely reversed the inhibition of CFTR by E-Syt3, while PtdC and PtdE had no effect. Phosphatidic acid (PA), a product of DAG hydrolysis, also failed to reverse the inhibitory effect of E-Syt3 (Appendix Fig. S8A). Importantly, unlike PtdSer, infusing the cells with the signaling lipid PI(4,5)$P_2$ that has higher negative charge than PtdSer only minimally affected inhibition by E-Syt3 (Fig. 3F).

Expression of E3SMPX together with E3C2C also reduced junctional PtdSer (Appendix Fig. S8B) and infusion of PtdSer, but not of PtdE, reversed the inhibition caused by E3SMPX and E1SMPX expressed with E3C2C (Appendix Fig. S9A) and by high level of E3SMP (Appendix Fig. S10A). Similar results were obtained when CFTR was inhibited by expression of the E3SMP and E3C2C (Fig. 3Hi–ii), where only PtdSer, but not PtdC and PtdE, reversed the inhibition. Similar studies were carried out with NBCe1-B. Figure 3Gi–iii,Ii–ii shows that PtdSer, but not PtdC, PtdE or PI(4,5)

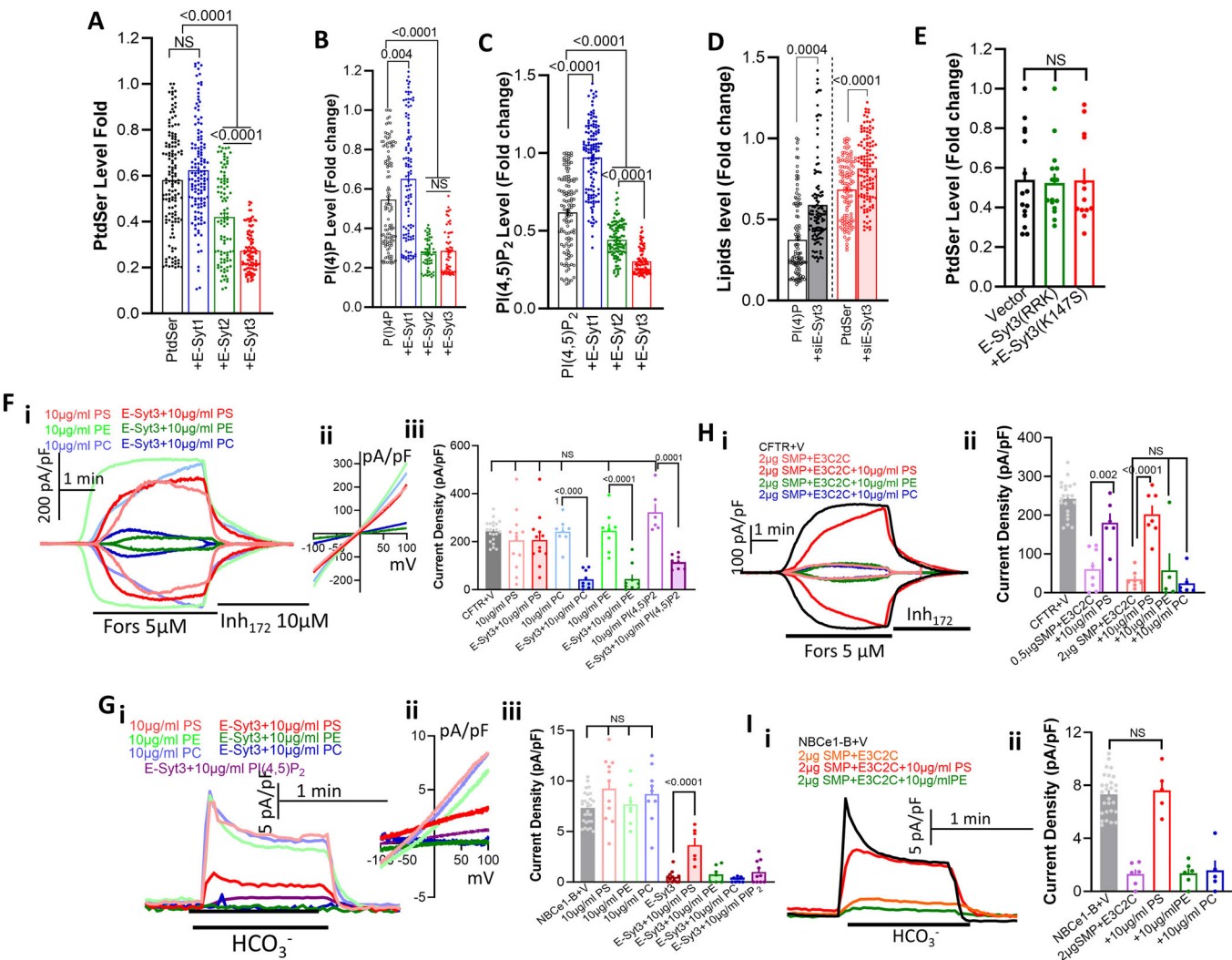

**Figure 3. Selective and localized reduction in plasma membrane PtdSer mediates E-Syt3 effects.**

(A–C) Effect of E-Syt1 (blue), E-Syt2 (green) and E-Syt3 (red) on steady-state junctional PtdSer (A), PI(4)P (B), and PI(4,5)P$_2$ (C) level. Results (mean ± s.e.m) are from six independent experiments for each condition and each dot represents one cell. (D, E) Effect of siE-Syt3 (D, n = 6), E-Syt3(R852/R853/K855A) and E-Syt3(K147S) (E, n = 3) on steady-state junctional PI(4)P and PtdSer level. Each dot represents one cell. (Fi–iii, Gi–ii) CFTR (F) and NBCe1-B (G) currents in HEK293T cells with (filled columns) or without E-Syt3 (open columns) and infused with PtdSer (red), PtdE (green), PtdC (blue) or PI(4,5)P$_2$ (purple). The results are from 3-4/6-13 T/C and are shown as mean ±s.e.m. (Hi–ii, Ii–ii) CFTR (H) and NBCe1-B (I) currents in HEK293T cells expressing SMP + E3C2C and infused with PtdSer (red), PtdE (green), PtdC (blue) or PI(4,5)P$_2$ (purple). Results for (H) are 2/5-8 T/C and for (I) are 2/5-6 T/C and are shown as mean ± s.e.m. All P values were determined by unpaired Student t test. Source data are available online for this figure.

P$_2$, reversed the inhibition of NBCe1-B by E-Syt3 or by the E3SMP + E3C2C. Appendix Figs. S9B and S10B show that PtdSer, but not PtdE, reversed the inhibition of NBCe1-B by E3SMPX + E3C2C and by high levels of E3SMP, respectively.

The overall findings clearly show that E-Syt3 acts through reduction in the E3C2C specified junctional PtdSer that can be rapidly and fully reversed by restoring PtdSer. The change in PtdSer is highly specific and localized in a subdomain accessed by E3C2C. Remarkably, similar reduction in junctional PtdSer by E-Syt2 (Fig. 3A) and massive PtdSer externalization (Appendix Fig. S7) had no effect or marginally activated CFTR.

To determine whether acute reduction in junctional PtdSer was sufficient to inhibit the transporters, first we tested the effect of

pharmacological reduction in general PM PtdSer using two independent inhibitors of PtdSer synthesis, Fendiline (Zhou and Hancock, 2018) and GW4869 (Gbotosho et al, 2014), to deplete PM PtdSer (Appendix Fig. S11). Preincubating cells with 25 µM Fendiline or 20 µM GW4869 for 30 min significantly inhibited CFTR (Appendix Fig. S11A–C) and NBCe1-B activity (Appendix Fig. S11D,E), that was largely reversed by acute supplementation of PtdSer. The specificity of Fendiline is shown by its lack of effect on the PtdSer-insensitive (see below) (Δ1-95)NBCe1-B (Appendix Fig. S11D). Second, the role of specific acute changes in PtdSer was determined by targeting the E3SMP to PM caveolae with the FRB/FKBP/rapamycin targeting system (Varnai et al, 2006). Treating cells expressing CFTR, FRB, E3SMP-FKBP with rapamycin was

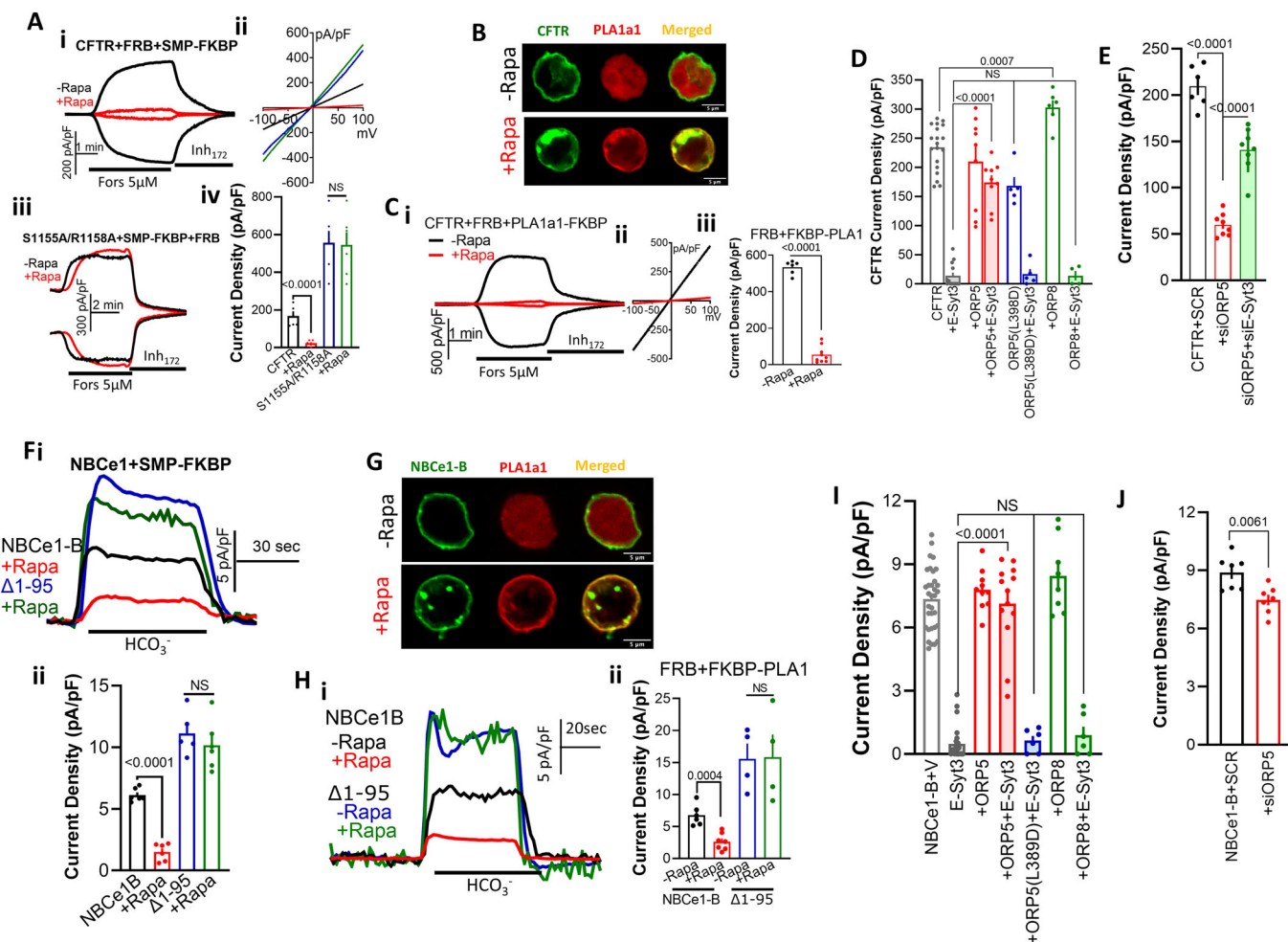

**Figure 4. Depletion and supplementation of junctional PtdSer regulate CFTR and NBCe1-B activity.**

(A) cells expressing CFTR (i) or CFTR(S1155A/R1158A) (iii) and FRB + SMP-FKBP were treated with buffer (black) or 0.5 μM rapamycin for 8 min (red) before stimulation with F/I, (ii) example I/Vs and (iv) summary. Results are from 2/6 T/C and are shown as mean ± s.e.m. (B, G) Localization of PLA1a1-FKBP before and after rapamycin treatment in cells expressing FRB and CFTR (B) or NBCe1-B (G). Results for (B) are 2-3/6-9 T/C and for (G) are 1-2/4-7 T/C and are shown as mean ± s.e.m. (Ci–iii) cells expressing CFTR and FRB+PLA1a1-FKBP were treated with buffer (black) or 0.5 μM rapamycin (red) for 8 min before stimulation with F/I. Results are from 3/6-9 T/C and are shown as mean ± s.e.m. (D) Current was measured in cells expressing CFTR (open columns) or CFTR + E-Syt3 (close columns) and ORP5 (red), ORP5(L389D) (blue) or ORP8 (green). Results are from ORP5(L389D) and ORP8 2/5-6 T/C, the rest 3-4/9-19 T/C and are shown as mean ± s.e.m. (E) CFTR current was measured in cells treated with scrambled (black), siORP5 (red), or siORP5+siE-Syt3. Results are from 3/6-8 T/C and are shown as mean ± s.e.m. (Fi–ii) Cells expressing NBCe1-B (black, red) or (Δ1-95)NBCe1-B (blue, green) and FRB + SMP-FKBP were treated with buffer (black) or 0.5 μM rapamycin for 8 min (red, green) before current measurement. Results are from 2/5-6 T/C and are shown as mean ± s.e.m. (Hi–ii) Cells expressing NBCe1-B or (Δ1-95)NBCe1-B and FRB+PLA1a1-FKBP were treated with buffer (blue) or 0.5 μM rapamycin (green) for 8 min before current measurement. Results are from 2/4-7 T/C and are shown as mean ± s.e.m. (I) Current was measured in cells expressing NBCe1-B (open columns) or NBCe1-B + E-Syt3 (close columns) and ORP5 (red), ORP5(L389D) (blue) or ORP8 (green). Results are from ORP5(L389D) and ORP8 2/6-8 T/C, the rest 4-7/10-33 T/C and are shown as mean ± s.e.m. (J) NBCe1-B current was measured in cells treated with scrambled (black) or siORP5 (red). Results are from 2/7 T/C and are shown as mean ± s.e.m. All P values were determined by unpaired Student t test. Source data are available online for this figure.

sufficient to inhibit CFTR activity (Fig. 4Ai,ii,iv), while the mutant E3SMP(K207E/I208E) when expressed with E3C2C had no effect on CFTR current (Appendix Fig. S10C). As a control, the constitutively active CFTR(S1155A/R1158A) (see below) was unaffected (Fig. 4Aiii–iv). Similar results were obtained with NBCe1-B, and as a control with (Δ1-95)NBCe1-B (Fig. 4Fi–ii). Third, independent evidence was obtained by targeting the mouse PtdSer-specific PLA1a1 that hydrolyzes PtdSer at the sn-2 position (Aoki et al, 2002) to the caveolar domain (Chung et al, 2023). Upon expression of RFP-PLA1a1-FKBP with FRB and CFTR (Fig. 4B) or NBCe1-B (Fig. 4G) RFP-PLA1a1-FKBP was cytoplasmic before the

addition of rapamycin. Treatment with 0.5 μM rapamycin transferred PLA1a1 to FRB at the PM that markedly inhibited CFTR (Fig. 4Ci–iii) and NBCe1-B (Fig. 4Hi–ii) activity to the same extent as E-Syt3.

## PtdSer transfer by ORP5 antagonizes the function of E-Syt3

The LTPs ORP5 and ORP8 exchange PM PI(4)P for ER PtdSer to control the junctional level of these lipids with ORP5 primarily increasing PM PtdSer and ORP8 primarily increasing PM PI(4)P

 

((Chung et al, 2015; Chung et al, 2023) and Appendix Fig. S12A,B). Furthermore, depletion of ORP5 markedly reduced junctional PtdSer and slightly increased PI(4)P, while depletion of ORP8 markedly reduced junctional PI(4)P and slightly increased PtdSer (Appendix Fig. S12A,B). Therefore, we tested whether lipid transfer by ORP5 and ORP8 affect the function of the transporters and their regulation by E-Syt3. Figure 4D,I show that ORP5 and ORP8 had no effect on CFTR and NBCe1-B basal current when expressed alone. However, ORP5 completely reversed the inhibition seen by E-Syt3 while ORP8 had no effect (Fig. 4D,I). Conversely, depletion of ORP5 significantly inhibited CFTR and NBCe1-B currents (Fig. 4E,J). Notably, the native E-Syt3 and ORP5 reciprocally affected CFTR current with depletion of E-Syt3 that increased PtdSer reversed the inhibition caused by depletion of ORP5 (Fig. 4E), and the effect of ORP5 on junctional PtdSer and PI(4)P (Appendix Fig. S12C,D). The effects of depleting (Appendix Fig. S12E,F) or expressing ORP5 (Appendix Fig. S12G,H) were not due to changes in CFTR and NBCe1-B levels at the TIRF field, but rather due to an effect on junctional lipids. Reversal of other essential functions of E-Syt3 by ORP5 are described below.

## E-Syt3 and PtdSer have no direct effect on CFTR and NBCe1-B conductance

Several experimental protocols were used to determine if E-Syt3 and PtdSer directly regulate CFTR and NBCe1-B conductance. This was not the case since pre-activation of the transporters prevented the inhibition by E-Syt3 and by hydrolysis of PtdSer. First, stimulating the cells with 0.2 μM forskolin starting at the time of transfecting the cells with CFTR and E-Syt3 (higher forskolin concentrations were toxic) largely prevented inhibition of the residual CFTR activity by E-Syt3 (Appendix Fig. S13Ai–iii). Second, incubating the cells with the CFTR potentiator VX-770 that increased spontaneous CFTR activity prevented inhibition by E-Syt3 (Appendix Fig. S13Bi–iii). Third, the constitutively active CFTR(E1371Q) mutant (Li et al, 2012) was not inhibited by E-Syt3 (Fig. 5Ai–iii). Fourth, targeting the E3SMP to the PM did not inhibit the constitutively active CFTR(S1155/R1158A) (Fig. 4Aiii). Fifth, the basal and forskolin-stimulated activity of CFTR(S1155/R1158A) is not inhibited by hydrolysis of junctional PtdSer with PLA1a1 (Fig. 5Bi–iii), indicating that the lack of inhibition by depletion of PtdSer is irrespective of the constitutively active CFTR mutant. Finally, hydrolysis of PtdSer after activation of CFTR slightly but significantly increased, rather than inhibiting CFTR activity (Appendix Fig. S13Ci–iii). NBCe1-B is also not directly inhibited by PtdSer depletion. As shown in Fig. 4F,H, the constitutively active (Δ1–95)NBCe1-B is not inhibited by E3SMP- and PLA1a1-mediated reduction in junctional PtdSer. In addition, Appendix Fig. S13D shows that PtdSer hydrolysis after activation of NBCe1-B by $HCO_3^-$ increased rather than inhibited NBCe1-B activity.

## The NBCe1-B and CFTR PtdSer sensors

The PtdSer-dependent inhibition of NBCe1-B and CFTR by E-Syt3 requires NBCe1-B and CFTR domains that can sense and bind PtdSer. The lack of inhibition by E-Syt3 and PtdSer hydrolysis of (Δ1–95) NBCe1-B suggested that the NBCe1-B PtdSer sensor is located at the AID. To test this, we assayed the activity of several additional NBCe1-B mutants and other NBC transporters that do not possess AIDs. Basal NBCe1-B activity is markedly stimulated by IRBIT which prevents the

inhibition of NBCe1-B by AID (Hong et al, 2013; Shirakabe et al, 2006). Appendix Fig. S13E shows that E-Syt3 inhibits the basal and the IRBIT-activated NBCe1-B. The NBCe1-B(S232A/S233A/S235/A) mutant is fully active in the presence and absence of IRBIT but possesses the AID (Vachel et al, 2018), and accordingly it was largely inhibited by E-Syt3. Moreover, E-Syt3 did not inhibit the NBCe1-A isoform that differs from NBCe1-B in only the first 40 residues and lacks autoinhibitory function, and the $Na^+/HCO_3^-$ transporter NBCe2C that does not possess AID (Appendix Fig. S13E). Thus, the inhibition of NBCe1-B by E-Syt3 and by PtdSer depletion requires an intact AID domain.

PtdSer is a negatively charged lipid and is likely detected by a stretch of positively charged residues, as is often found for the phosphoinositides (Gokhale, 2013). Therefore, we tested if mutating the positively charged residues within the AID affect inhibition by E-Syt3. Previously, we proposed that NBCe1-B R42, R43 and R44 may affect regulation of NBCe1-B by PI(4,5)P₂ (Hong et al, 2013). However, Fig. 5C shows that mutation of these residues and even of only R44 eliminated inhibition of NBCe1-B by E-Syt3, while NBCe1-B(R42A/R43A) was fully inhibited by E-Syt3. The IAD is the NBCe1-B IRBIT binding domain (Shirakabe et al, 2006) and deletion of the AID and the R44A mutation markedly reduced the NBCe1-B-IRBIT FRET signal (Fig. 5D). Interestingly, the (Δ1–95)NBCe1-B and NBCe1-B(R44A) mutants that were not inhibited by E-Syt3 showed normal Co-IP with E-Syt3 (Fig. 5E) and even enhanced FRET with E-Syt3 (Fig. 5F), suggesting that the AID has no role in targeting NBCe1-B to the domain to where E3C2C targets E-Syt3. To determine whether the AID binds PtdSer, we purified His-tagged NBCe1-B(1-95) expressed in *E. coli* and assayed its binding to beads in buffer (empty) or beads impregnated with PtdSer or PtdC. Figure 5G shows that beads containing PtdSer, but not PtdC, pulled down the NBCe1-B(1–95). As an additional control, we generated His-tagged NBCe1-B(1–95) R44A mutant and Fig. 5H shows that the mutant lost PtdSer binding. Thus, the results in Fig. 5C–H suggest that the AID is the PtdSer sensing domain of NBCe1-B.

The structures of inactive CFTR and of ATP and potentiator-activated CFTR (Fiedorczuk and Chen, 2022; Liu et al, 2017; Zhang et al, 2017) suggest that the CFTR, R domain functions as an AID. Therefore, we tested whether the R domain is the CFTR PtdSer sensor: first, we mutated several residues in CFTR transmembrane domains that contact the R domain (Appendix Fig. S14A). The effect of the mutations on current density and inhibition by E-Syt3 are summarized in Appendix Fig. S14B–D. The K978C mutant that was reported before (Wang et al, 2010) and the S1155A/R1158A mutant are constitutively active with higher maximal current than the forskolin-stimulated CFTR and were not inhibited by E-Syt3. The Q1042A/E1046A and S1049A/F1052A mutants had normal or reduced activity but were partially inhibited by E-Syt3 (Appendix Fig. S14C,D). All the mutants tested showed similar Co-IP with E-Syt3 (Appendix Fig. S14E). However, although the R domain does appear to function as AID, it is unlikely to be the CFTR PtdSer sensor since mutating the 3 positively charged arginine in the R domain (Appendix Fig. S15A) and deletion of the entire R domain (Appendix Fig. S15B) were still largely inhibited by E-Syt3. The ATPase activity of CFTR NBD1 is stabilized by PtdSer (Hildebrandt et al, 2017). However, NBD1 is far from the PM (Fiedorczuk and Chen, 2022; Liu et al, 2017). Moreover, mutating lysine K612, K613 and K615 that are at NBD1 exposed surface (Appendix Fig. S16A) had no effect on the inhibition by E-Syt3 (Appendix Fig. S16B).

 

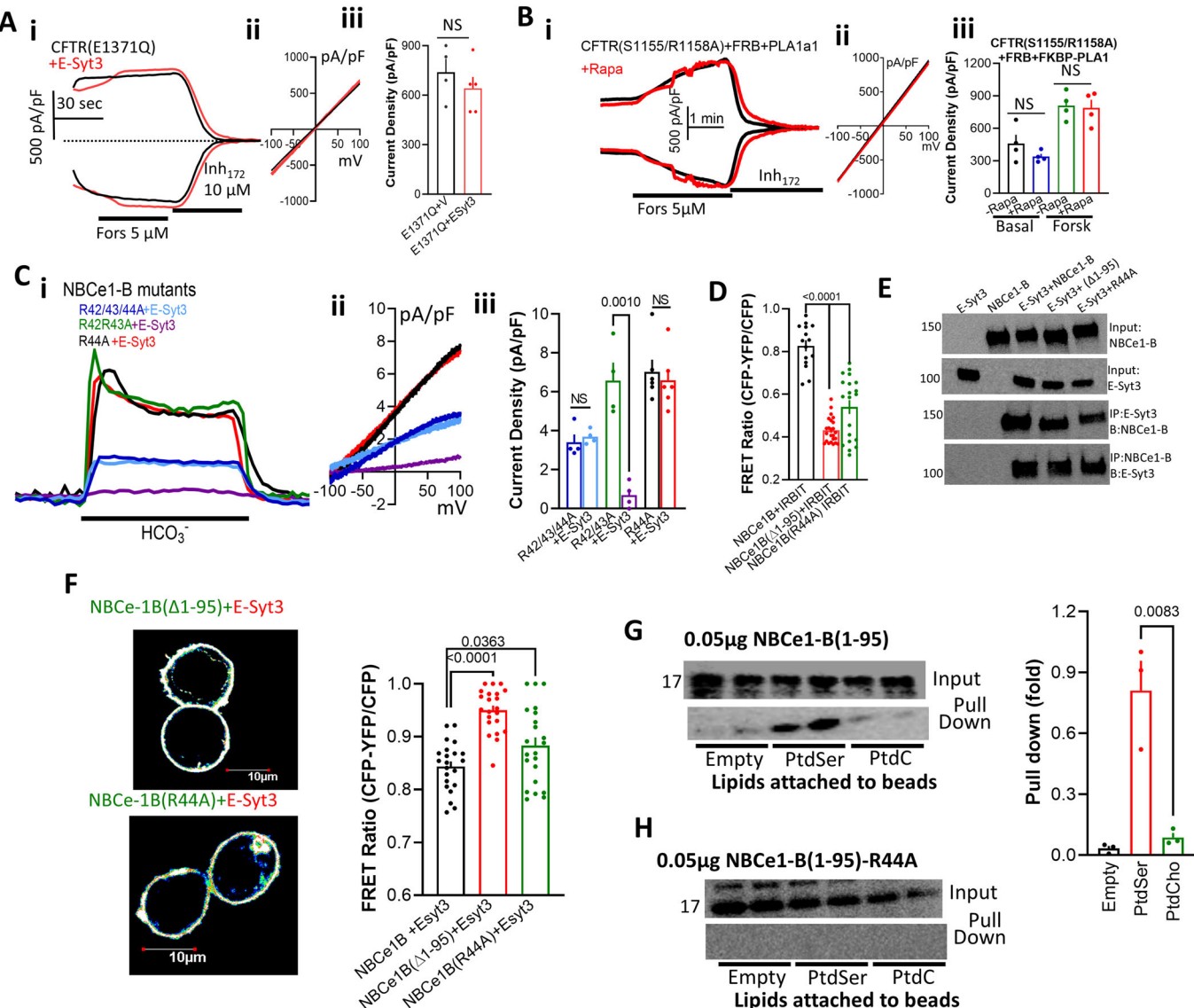

**Figure 5. PtdSer does not affect CFTR, and NBCe1-B pore properties and the NBCe1-B PtdSer sensor.**

(A, B) Current was measured in cells expressing vector (black) or E-Syt3 (red) and the constitutively active CFTR(E1371Q) (A) or CFTR(S1155A/R1158A) (B). Results are from (A) 2/4-5 T/C and (B) 2/4 T/C and are shown as mean ± s.e.m. (Ci–iii) Current measured with cells expressing NBCe1-B(R42A/R43A/R44A), NBCe1-B(R42A/R43A) or NBCe1-B(R44A) with vector or E-Syt3, as indicated. and are shown as mean ± s.e.m. (D) FRET was measured between GFP-tagged NBCe1-B (black), NBCe1-B(Δ1-95) (red), NBCe1-B(R44A) (green) and IRBIT-CFP. Results are from 4/15-24 T/C and are shown as mean ± s.e.m. (E) Cells transfected with the indicated combinations of NBCe1-B mutants and E-Syt3 were used to determine their reciprocal Co-IP. (F) FRET measured between GFP-tagged NBCe1-B (black) (Δ1-95)NBCe1-B (red) or NBCe1-B(R44A) and E-Syt3-mCherry with example images and summary (columns). Results are from 4/22 T/C and are shown as mean ± s.e.m. (G) Blots of the His-NBCe1-B AID (first 95 residues of NBCe1-B) that were pulled down with empty beads and beads impregnated with PtdSer or PtdC and probed with anti-His. The upper blot is the input, and the lower the pulled AID, and the columns are the summary (n = 3, mean ± s.e.m.) (H) The same as (G), but with the AID mutant (Δ1-95)NBCe1-B (R44A) (n = 3, mean ± s.e.m.) All *P* values were determined by unpaired Student *t* test. Source data are available online for this figure.

A highly positively charged domain that is close to the PM inner surface and therefore to PtdSer is the CFTR lasso domain ((Liu et al, 2017; Zhang et al, 2017) and Fig. 6A). We mutated clusters and combinations of positively charged lasso residues. The summaries in Fig. 6B, the example traces and I/Vs in Appendix Fig. S16C–H show that none of the mutants increased basal CFTR activity, although CFTR(R25/K26A) and CFTR(K64/K65A) markedly increased forskolin-stimulated CFTR current. Except for CFTR(R29/R31A), all mutants partially reduced inhibiting by E-Syt3. The most interesting stretch was K52/R55/R59 that are surface exposed in the lasso helix 2 that is perpendicular to the TMD (magenta residues in Fig. 6A). The CFTR(K52A/R55A/R59A) mutant is completely resistant to inhibiting by E-Syt3 (Fig. 6Ci–iii) and to hydrolysis of PtdSer by PLA1a1 (Fig. 6Di–iii). We generated and purified GST-Lasso domain and used it for PtdSer binding. Figure 6E–G shows that the Lasso domain binds PtdSer but not or very little PtdC and PI(4,5)P2. Mutating K52A/R55A/R59A of the lasso domain eliminated PtdSer binding (Fig. 6H). Thus, it appears that the Lasso domain is the CFTR PtdSer sensor.

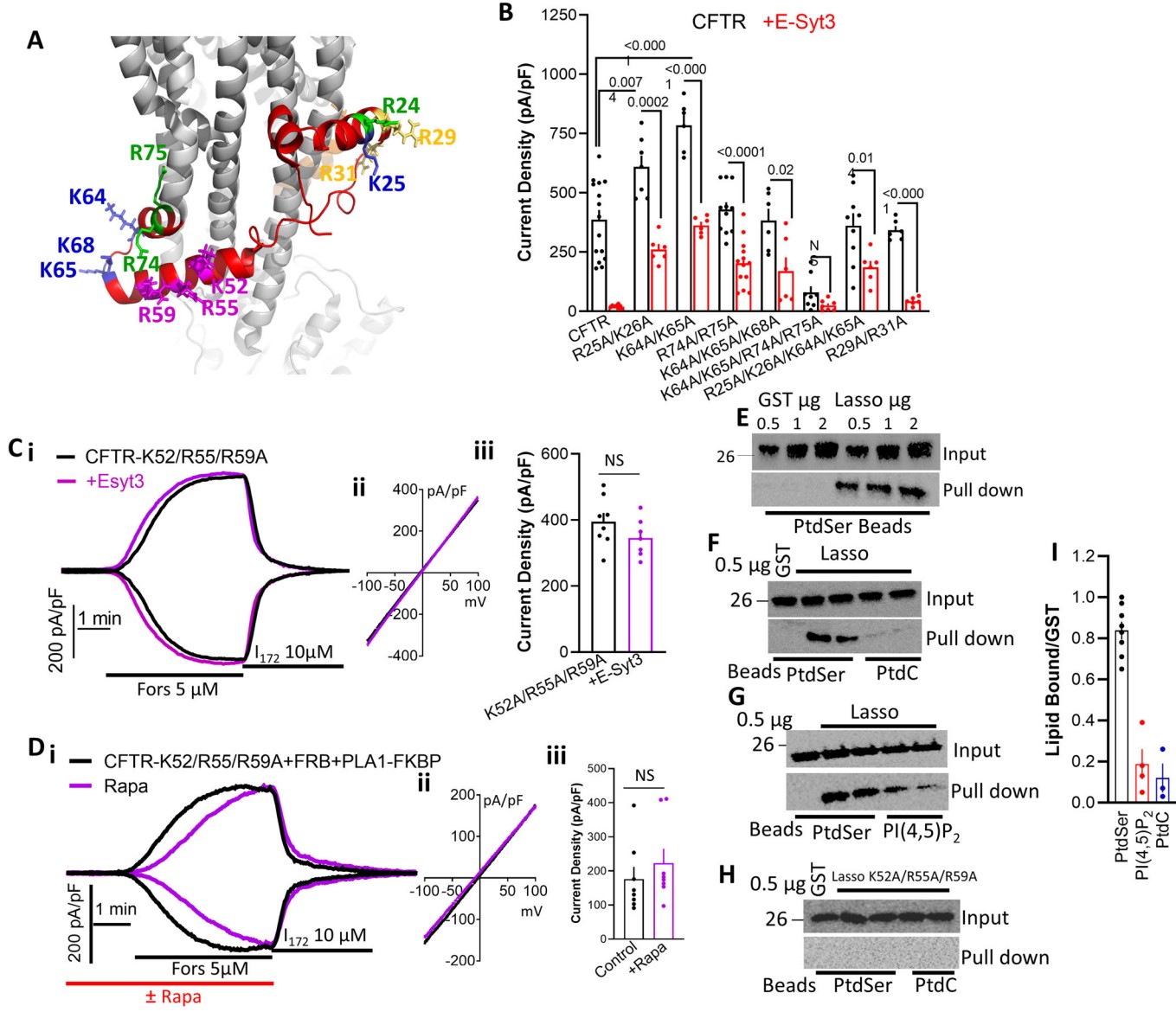

**Figure 6. The CFTR Lasso domain is the CFTR PtdSer sensor.**

(A) The structure of CFTR with the lasso domain shown in red and the mutated residues shown as sticks. (B) Current density in cells expressing the indicated CFTR mutants with either empty vector (black) or E-Syt3 (red). Example traces and I/V are given in Appendix Fig. S16. Results are from 2-4/6-13 T/C and are shown as mean ± s.e.m. (Ci–iii, Di–iii) Current was measured in cells expressing CFTR(K52A/R55A/R59A) without (black) and with E-Syt3 (purple) (C) or CFTR(K52A/R55A/R59A) + FRB+PLA1a1-FKBP (D) and treated with buffer (black) or 0.5 μM rapamycin for 8 min (purple). Results for (C) are from 2/6-7 T/C and for (D) are from 2/8 and are shown as mean ± s.e.m. All P values were determined by unpaired Student t test. (E–G, I) Blots of the indicated μg of GST (control n = 3) or GST-Lasso that were pulled down with beads impregnated with PtdSer (E, I, n = 8), with PtdSer or PtdC (F, I, n = 3) or with PtdSer or PI(4,5)P2 (G, I, n = 3). The upper blots are the input, and the lower blots are the pulled GST-Lasso. Results are given as mean ± s.e.m. (H) The same as (F), but with the Lasso mutant K52A/R55A/R59A. Source data are available online for this figure.

## E-Syt3 and PtdSer are required for protein complexes assembly and cAMP signal transduction, and are reversed by ORP5

The lack of PtdSer effect on CFTR and NBCe1-B conductance raised the question of how PtdSer exerts its regulation. CFTR is a cAMP-regulated channel and phosphorylation of the R domain by the PKA catalytic subunit mediates activation of CFTR (Csanady et al, 2019). To activate CFTR, cAMP needs to bind to the AKAP-

associated PKA regulatory subunit to release the PKAc catalytic subunit that can now phosphorylate the CFTR R domain (Omar and Scott, 2020). This requires the cAMP/PKA signaling pathways and CFTR to be present in a complex with AKAP and PKAc. Therefore, we tested whether E-Syt3 affects the AKAP-PKAc and CFTR-AKAP-PKAc complex and whether formation of the complex requires PtdSer. AKAP79 (Also known as AKAP5) is a prominent AKAP in epithelia, and we determined its association with CFTR, both by FRET and by Co-IP. Figure 7A,B shows a

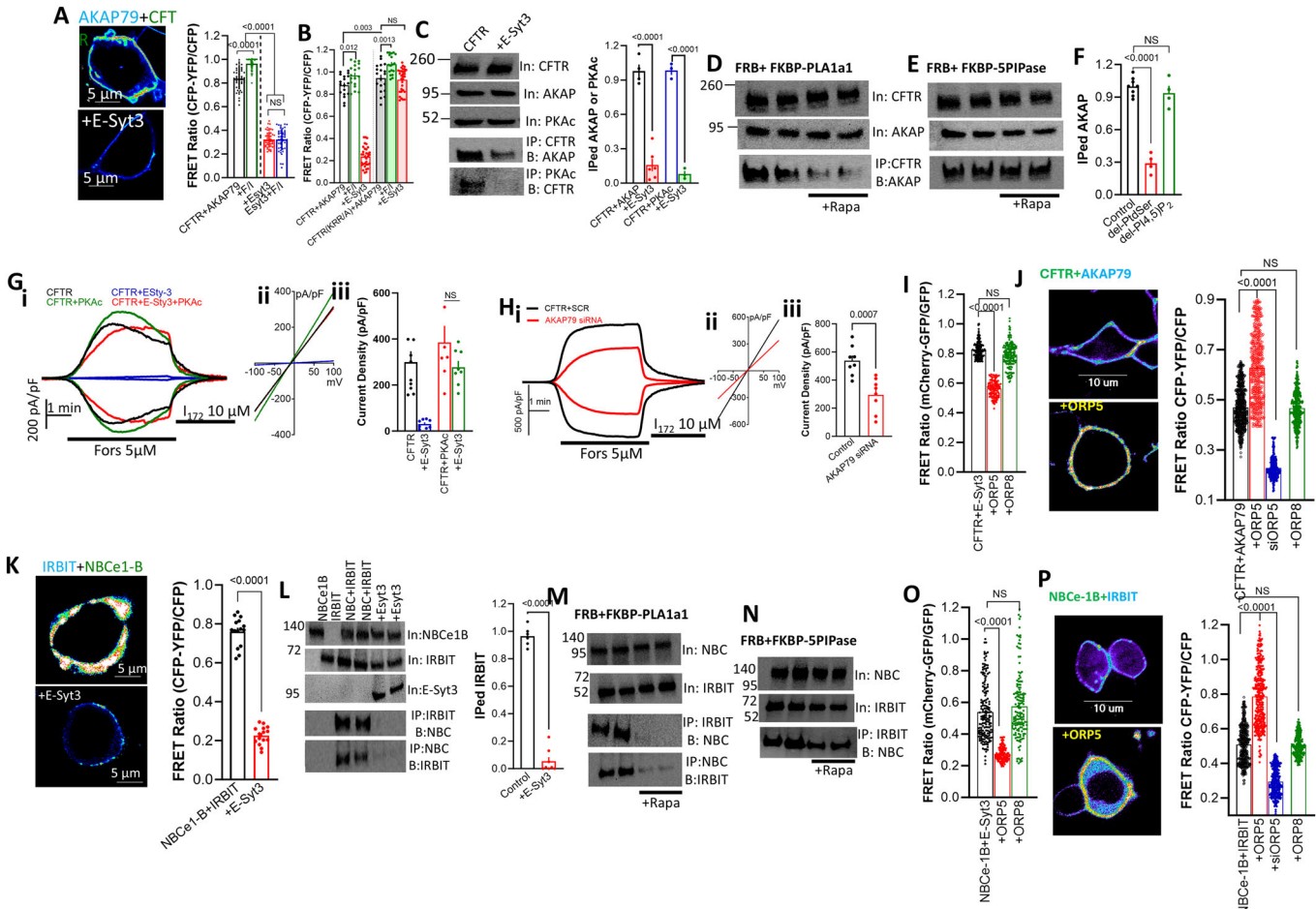

**Figure 7. PtdSer accessed by E-Syt3 and ORP5 regulates signaling complexes formation.**

(A) FRET was measured between CFTR-GFP and AKAP79-mTurqoise in the absence and presence of E-Syt3. Example images and summary (columns) of measurements with CFTR (black) simulated with F/I (green) and CFTR + E-Syt3 (red) stimulated with F/I (blue). Results are from 10/50 T/C and are shown as mean ± s.e.m. (B) FRET was measured between CFTR-AKAP79 and between the mutant CFTR(KRR/AAA)-AKAP79 in the presence and absence of E-Syt3, as indicated. Results are from 4-7/18-33 T/C and are shown as mean ± s.e.m. (C) Cells transfected with CFTR alone or CFTR + E-Syt3 (n = 6) were used to measure Co-IP of CFTR with the native AKAP79 (n = 4) or the native PKA catalytic subunit (PKAc, n = 3). (Left) example blots and (right) the averages. Results are shown as mean ± s.e.m. (D–F) Cells transfected with CFTR + FRB+PLA1a1-FKBP (to deplete PtdSer) or FKBP-5PIPase (to deplete PI(4,5)P$_2$) and treated with buffer or 0.5 µM rapamycin for 8 min before preparation of lysates that were used to determine Co-IP of CFTR and the native AKAP79. (D) effect of PtdSer depletion (n = 4), (E) effect of PI(4,5)P$_2$ depletion (n = 4) and (F) the summary as mean ± s.e.m. (Gi-iii) Cells expressing CFTR (black, green) or CFTR + E-Syt3 (Blue, red) were infused with 400 U/ml PKAc for 8 min (CFTR green, CFTR + E-Syt3 red) before stimulation. Results are from 3/2-10 T/C and are shown as mean ± s.e.m. (Hi-iii) Cells treated with scrambled (black) or AKAP79 siRNA were transfected with CFTR, and the current in response to F/I was measured. Results are from 2/8 T/C and are shown as mean ± s.e.m. (I) FRET between CFTR-GFP and E-Syt3-mCherry was measured in the presence of ORP5 and ORP8. Results are from 15-21/129-165 T/C and are shown as mean ± s.e.m. (J) FRET was measured between CFTR-GFP and AKAP79-mTurqoise in the absence of ORP5 (red), ORP8 (green), and cells depleted of ORP5 (blue). Results are from 6-15/32-60 T/C and are shown as mean ± s.e.m. (K) FRET between NBCe1-B-GFP and its activator IRBIT-CFP was measured in the absence and presence of E-Syt3. Shown are example images and the summary 3/15 T/C and are shown as mean ± s.e.m.) (L) Cells expressing NBCe1-B and IRBIT with (n = 8) and without E-Syt3 (n = 6) in the indicated combination were used to determine the Co-IP between NBCe1-B and IRBIT. Shown are example blots and the summary as mean ± s.e.m. (M, N) Cells transfected with NBCe1-B + IRBIT + FRB and either PLA1a1-FKBP (M) or FKBP-5PIPase (N) were treated with buffer or 0.5 µM rapamycin for 8 min before preparation of lysates that were used to determine Co-IP of NBCe1-B and IRBIT. (O) FRET between NBCe1-B-GFP and E-Syt3-mChrerry was measured in the presence of ORP5 and ORP8. Results are from 16-26/126-172 T/C and are shown as mean ± s.e.m. (P) FRET was measured between NBCe1-B-GFP and IRBIT-CFP in the absence of ORP5 (red), ORP8 (green) and cells depleted of ORP5 (blue). Results are from 5-7/25-45 T/C and are shown as mean ± s.e.m. All P values were determined by unpaired Student t test. Source data are available online for this figure.

prominent FRET signal between CFTR-GFP and AKAP79-mTurqoise2 that was significantly enhanced by stimulation with forskolin. Notably, E-Syt3 strongly inhibited the FRET signal that could not be rescued by forskolin stimulation. Moreover, mutating the PtdSer binding sensor site further increased the CFTR-AKAP79 association and completely prevented inhibition of CFTR-AKAP79 interaction by E-Syt3 (Fig. 7B). Similar results were obtained by

assaying Co-IP between CFTR and AKAP79 (Fig. 7C). In addition, E-Syt3 disrupted transduction by the cAMP signaling pathway as evident from disruption of the interaction between CFTR and the catalytic subunit of PKA (PKAc) (Fig. 7C).

Of particular importance, hydrolysis of PtdSer by PLA1a1 was sufficient to disrupt transduction by the cAMP pathway, as indicated by dissociation of the CFTR-AKAP79 complex

 

(Fig. 7D,F). This was specific to PtdSer, since depletion of $PI(4,5)P_2$ by the FRB/FKBP method had no effect (Fig. 7E,F), indicating that PtdSer is essential for the formation of the CFTR-AKAP79-PKAc complex and activation of CFTR. To obtain further evidence for altered transduction by the cAMP/PKA pathway, it is necessary to show that PKA can still phosphorylate and activate CFTR. Therefore, we tested if supplying the cells with excess PKAc that directly activates CFTR (Liu et al, 2017) can reverse the inhibition by E-Syt3. Indeed, infusing the cells with recombinant PKAc through the patch pipette fully activated CFTR in cells expressing CFTR and E-Syt3 (Fig. 7Gi–iii). In addition, the knockdown of AKAP79 should reduce activation of CFTR by forskolin stimulation. The siAKAP79 used reduced both the level of AKAP79 protein (Appendix Fig. S18A) and CFTR current by about 50% (Fig. 7Hi–iii). Thus, PtdSer is required for both cAMP complex formation and signal transduction.

To examine how ORP5 may reverse the effect of E-Syt3, we measured the effect of ORP5 on the interaction with E-Syt3 and the formation of the complexes. Notably, ORP5, but not ORP8, significantly reduced the FRET between CFTR and E-Syt3 (Fig. 7I). Accordingly, FRET measurements showed that ORP5, but not ORP8, increased the interaction of CFTR with AKAP79, while depletion of ORP5 prominently reduced the interaction of CFTR with APAK79 (Fig. 7J), accounting for the reversal of the effects of E-Syt3 on CFTR activity by ORP5 (Fig. 4).

The effects of PKAc and AKAP79 were specific for CFTR since PKAc did not reverse the inhibition of NBCe1-B by E-Syt3, and knockdown of AKAP79 had no effect on NBCe1-B activity (Appendix Fig. S17A,Bi–iii). Since activation of NBCe1-B requires interaction of NBCe1-B with IRBIT (Hong et al, 2013; Shirakabe et al, 2006) we tested the effect of E-Syt3 on the interaction of NBCe1-B-GFP with IRBIT-CFP. Figure 7K,L shows prominent FRET and Co-IP between NBCe1-B-GFP and IRBIT-CFP that is markedly reduced by E-Syt3. Significantly, hydrolysis of PM PtdSer was sufficient to dissociate the NBCe1-B-IRBIT complex (Fig. 7M), while hydrolysis of PM $PI(4,5)P_2$ had minimal effect (Fig. 7N). The interaction of NBCe1-B with E-Syt3 was strongly reduced by ORP5 but was not affected by ORP8 (Fig. 7O). In addition, ORP5, but not ORP8, increased the interaction of NBCe1-B with IRBIT, while depletion of ORP5 prominently reduced the interaction of NBCe1-B with IRBIT (Fig. 7P). Together, the effects of ORP5 on E-Syt3 interaction and function suggest that E-Syt3 and ORP5 likely access the same junctional domain to determine its PtdSer levels, playing a crucial regulatory role in the function of the $HCO_3^-$ transporters CFTR and NBCe1-B.

## The physiological function of E-Syt3

Deletion of individual or all the E-Syts in mice had no apparent phenotype (Sclip et al, 2016). However, compensatory and adaptive mechanism(s) may have masked their physiological role. Moreover, subtle regulatory mechanisms that primarily function during stressed or pathological states are likely to be missed in knockout mice. Considering the plethora of the E-Syts cellular effects (Benavides and Giraudo, 2024; Maleth et al, 2014; Pan et al, 2023; Saheki and De Camilli, 2017; Wang et al, 2023), their function may show up better when increased or deleted acutely. Therefore, we used short-term and acute changes in E-Syt3 to reveal its physiological function. To establish the in vivo role of E-Syt3, we followed salivary gland function

(Fig. 8). Appendix Fig. S18B,C shows the relative mRNA level of the three E-Syts in HEK cells and submandibular glands (SMG). Interestingly, in both cell types E-Syt3 is expressed at a significantly lower level than E-Syt1 and in SMG E-Syt2 is also expressed at a lower level than E-Syt1, and yet only E-Syt3 affects the regulation of CFTR and NBCe1-B. Salivary gland ducts express and use the luminal membrane (LM) CFTR and the basolateral membrane (BLM) NBCe1-B to absorb the $Na^+$ and $Cl^-$ from the fluid secreted by the acini (Lee et al, 2012) (see model in Fig. 9F). Figure 8A shows that the native SMG CFTR and E-Syt3 co-precipitate and Fig. 8B shows that the native E-Syt3 is expressed both in the BLM (marked by yellow arrowheads) and the LM (marked by red arrowheads) regions of the intralobular ducts. The level of ductal E-Syt3 was increased by expression of E-Syt3-myc with the aid of a viral vector (Appendix Fig. S18D,E) delivered through the opening of the duct to the oral cavity (see image in (Hong et al, 2015)). E-Syt3-myc co-precipitated with the native CFTR (Fig. 8C) and showed the same ductal expression pattern as the native E-Syt3 (Fig. 8D). After 8 days of infecting the salivary glands by infusion of Adeno-GFP (control) or Adeno-E-Syt3-myc had no effect on the mice weight (Fig. 8E) or saliva volume (Fig. 8F), that is determined primarily by acinar cells (Lee et al, 2012; Melvin et al, 2005). Notably, increasing E-Syt3 expression increased saliva $Cl^-$ (Fig. 8G) and $Na^+$ (Fig. 8H) content, indicating impaired ductal NaCl absorption that is mediated by CFTR and ENaC. The reciprocal effect was obtained by depletion of E-Syt3 by infusing the glands with E-Syt3 siRNA (Fig. 8I), which reduced expression of E-Syt3 in the ducts (Fig. 8J). Treatment with siE-Syt3 had no effect on the mice body weight (Fig. 8K) and the volume of secreted saliva (Fig. 8L), but reduced saliva $Cl^-$ (Fig. 8M) and $Na^+$ (Fig. 8N) content, indicative of increased NaCl absorption by the salivary ducts.

To extend the physiological role of E-Syt3 observed in salivary glands to another tissue and used additional assays, we depleted E-Syt3 in the isolated sealed pancreatic ducts and measured $Cl^-$ fluxes and fluid secretion (Hong et al, 2015; Park et al, 2013; Zeng et al, 2017). CFTR-dependent $Cl^-$ fluxes were measured with the $Cl^-$−sensitive dye MQAE by incubating the duct in media in which $Cl^-$ was replaced with $NO_3^-$ to initiate $Cl^-/NO_3^-$ exchange by CFTR (Molnar et al, 2020; Park et al, 2013). Treating the sealed duct with control (siGLOGreen) and siE-Syt3 showed that depletion of E-Syt3 increased the CFTR-mediated $Cl^-$ fluxes (Fig. 9A,B). Unlike the salivary gland duct, the pancreatic duct secretes most of the pancreatic juice fluid (Lee et al, 2012). Therefore, we measured fluid secretion into sealed intralobular ducts microdissected from the mouse pancreas and treated with scrambled siRNA or siE-Syt3 in primary culture. The ducts spontaneously seal within 24–48 h in culture, and fluid secretion is recorded by monitoring images of ductal lumen expansion as they fill with fluid. The images in Fig. 9C show example of such a duct, and Fig. 9D shows that depletion of E-Syt3 increases the rate and extent of fluid secretion stimulated with the submaximal concentration of 1 μM forskolin. The averages in Fig. 9E show that the increased secretion was significant within the first 5 min of stimulation and remained so after 40 min of stimulation with forskolin. The combined results in Figs. 8 and 9 demonstrate the physiological role of E-Syt3 in epithelial fluid and electrolyte secretion.

## Discussion

A prominent mode of regulation of plasma membrane (PM) ion channels and transporters is by various lipids (Schmidpeter et al,

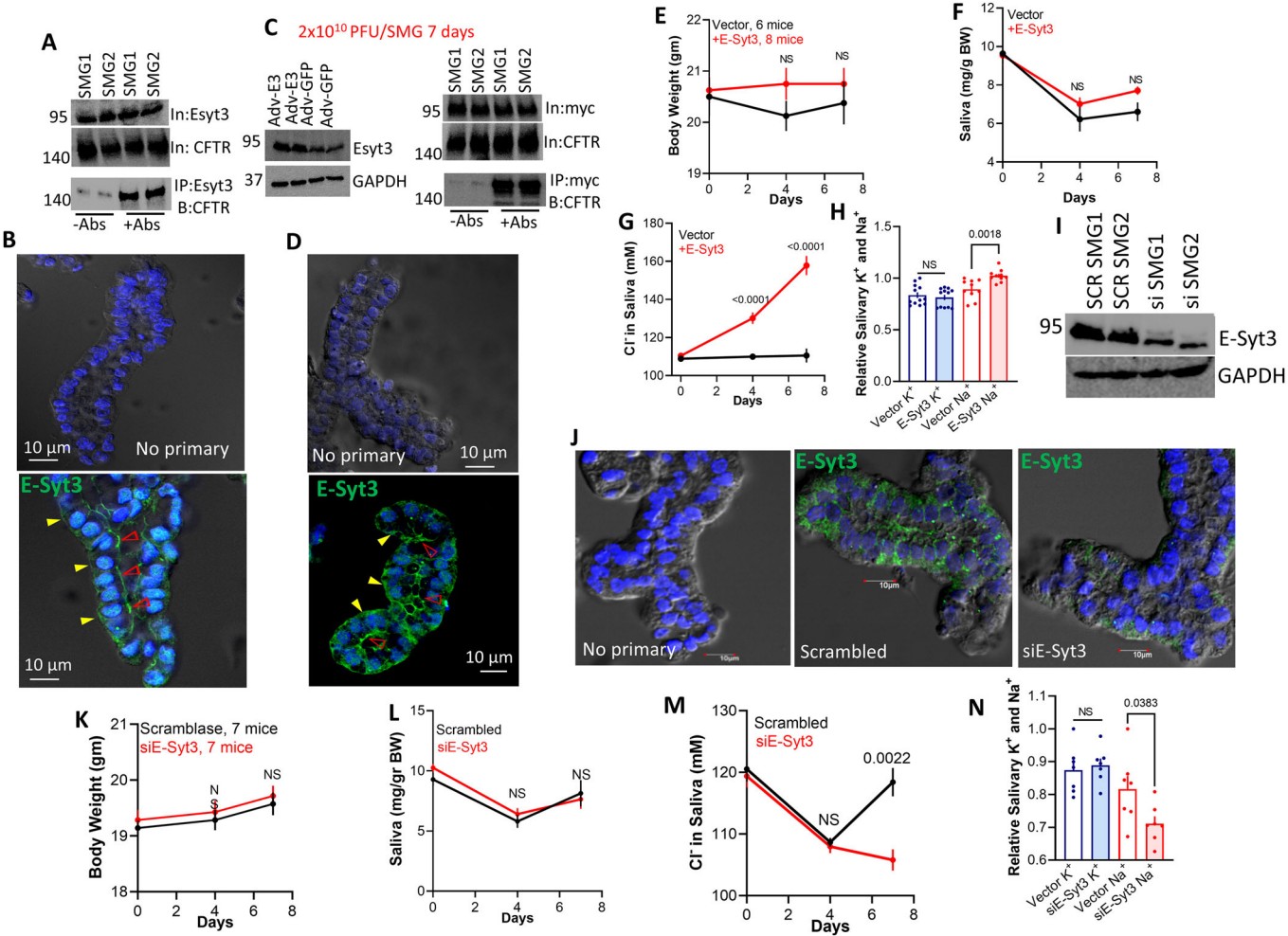

**Figure 8. Transgenic overexpression and siRNA depletion of E-Syt3 in mouse salivary glands alter ductal epithelial ion transport.**

(A, B) Submandibular glands (SMG) from two mice were used to measure Co-IP of the native CFTR and E-Syt3 (A) and to determine the basolateral and luminal localization of native E-Syt3 (B). In (B) are example images with no primary antibodies as control, green is E-Syt3, and blue is DAPI. In (B, D), the basolateral membrane is marked with yellow arrowheads and the luminal membrane with red arrowheads. (C, D) The mice SMG were infected with $2 \times 10^{10}$ PFU Adv-GFP (control) or Adv-E-Syt3-Myc by retrograde ductal infusion through the opening to the oral cavity. After 4 days the SMG were removed from two mice and used to measure Co-IP of the native CFTR and E-Syt3-Myc (C) and E-Syt3 localization (D). (E–H) The mice SMG were infected with GFP ($n = 6$, control, black) or E-Syt3-Myc ($n = 8$, red) and their weight measured 4- and 7 days post infection (E), while collecting saliva in response to stimulation with 0.5 mg/Kg pilocarpine (F). Saliva from the mice were analyzed for $Cl^-$ (G), $Na^+$ (H) and $K^+$ (H) concentrations. Results in all panels are shown as mean ± s.e.m. Each dop represent one mouse (I, J) The mice SMG were infused with scrambled or E-Syt3 siRNA and SMG from two mice were used to determine the level of E-Syt3 (I) and E-Syt3 localization in the glands (J) 7 days post treatment. (K–N) SMG were infused with scrambled ($n = 7$, black) or E-Syt3 ($n = 7$, red) siRNA and their body weight measured 4 and 7days post- siRNA infusion (K), the saliva was collected in response to stimulation with pilocarpine (L) and used to evaluate salivary $Cl^-$ (M), $Na^+$ and $K^+$ (N). Results in all panels are shown as mean ± s.e.m. Each dot represents one mouse. All $P$ values were determined by unpaired Student $t$ test. Source data are available online for this figure.

2022; Thompson and Baenziger, 2020). Lipids can function as signaling molecules that directly interact with transporters to modulate their function. Lipids also determine the transporters environment to affect their structure to indirectly regulate their function (Levental and Lyman, 2023; Renard and Byrne, 2021). The most prominent signaling lipids are the phosphatidylinositides, in particular PI(4,5)P$_2$ that regulates the activity of many channels and transporters (Dickson and Hille, 2019; Renard and Byrne, 2021; Thompson and Baenziger, 2020). The phosphatidylinositides are enriched in specific membrane domains, such as the ER/PM junctions and caveolae (Kefauver et al, 2024; Maleth et al, 2014; Myeong et al, 2021), to form specific lipid domains where ion

channels and transporter's function. Other important PM lipids are the phospholipids PtdC, PtdE and PtdSer and cholesterol. Cholesterol is an important structural lipid that determines the physical state of cellular membranes (Molugu and Brown, 2019), while very little is known about the role of the other phospholipids in regulation of ion transporters and cell signaling and transduction. Here, we report on a new role of PtdSer in the assembly of signaling complexes and signal transduction to regulate ion channels and transporters in precise and restricted ER/PM junctions.

PtdSer is synthesized in the ER and is distributed to other membrane compartments, primarily by lipid transfer proteins

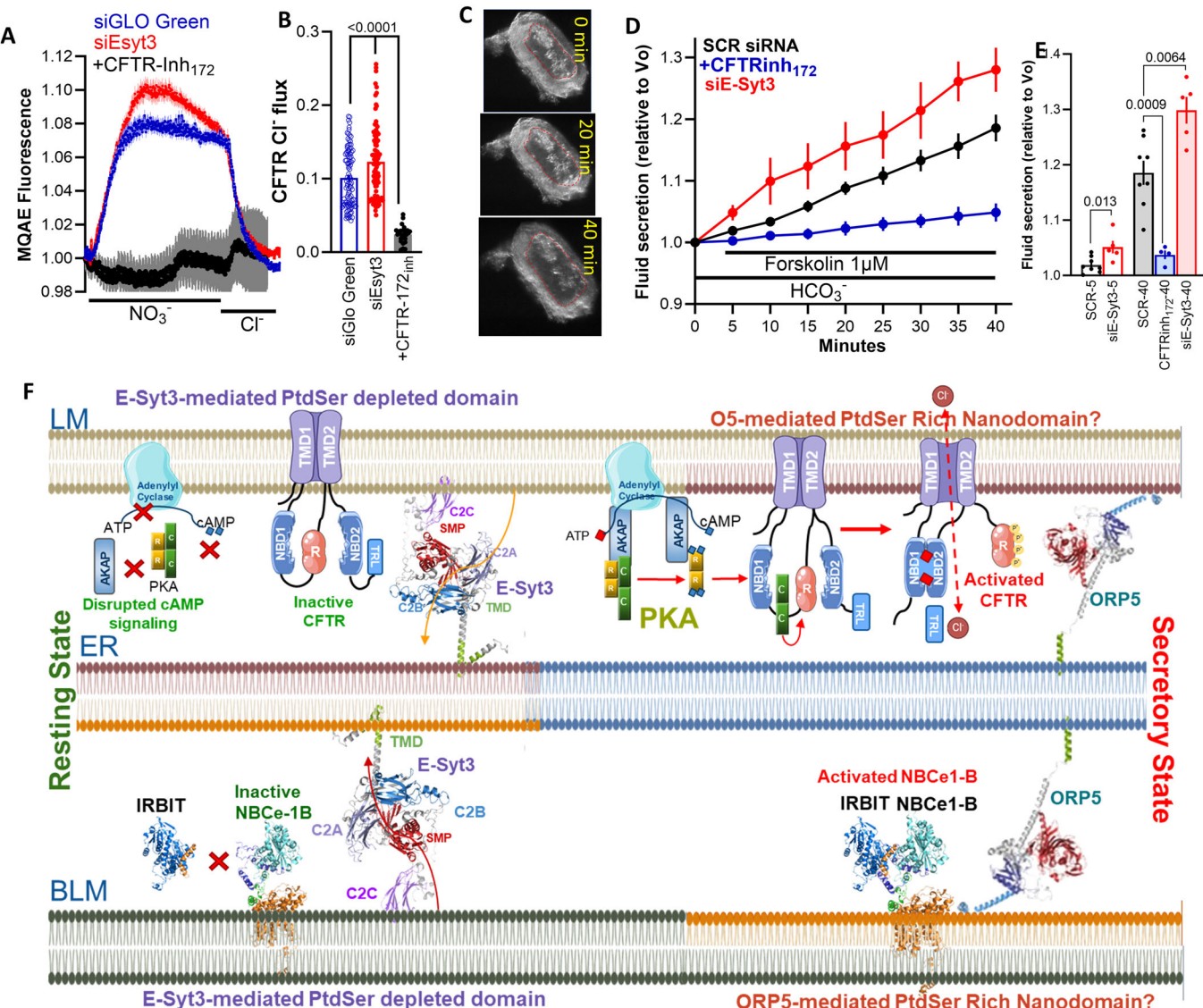

**Figure 9. Cl⁻ fluxes and fluid secretion by sealed mouse pancreatic ducts depleted of E-Syt3 and a model of E-Syt3-regulated epithelial fluid and HCO₃⁻ secretion.**

(A, B) Intracellular Cl⁻ was measured by MQAE fluorescence in pancreatic ductal fragments that were collected from three mice of each condition and treated with SCR siRNA (20 ducts), inhibited with 10 μM CFTRinh₁₂₇ (8 ducts) or treated with siE-Syt3 (17 ducts) and used to collect 104, 26, and 89 ROIs, respectively. Perfusing the duct with NO₃⁻ resulted in fluorescence increase due to Cl⁻ efflux by NO₃⁻$_o$/Cl⁻$_{in}$ exchange, and re-addition of Cl⁻ resulted in fluorescence decrease due to Cl⁻ influx. (A) Are average traces and (B) represents the maximal change of MQAE fluorescence upon removal of Cl⁻$_{out}$ in the indicated groups. Results are shown as mean ± s.e.m. (C–E) Ducts obtained from 4 mice were treated with scrambled siRNA (black, 8 ducts), and inhibited with 15 μM CFTRinh₁₂₇ (blue, 4 ducts) or treated with siE-Syt3 (red, 5 ducts) were used to measure fluid secretion. (C) Shows example images captured at 0, 5 and 40 min of a sealed duct treated with siE-Syt3 and stimulated with forskolin, (D) Is the time course of fluid secretion into sealed ducts stimulated by 1 μM forskolin, and (E) shows fluid secretion by the individual ducts at 5 and 40 min. Results are shown as mean ± s.e.m. All *P* values were determined by unpaired Student *t* test. (F) The model shows the localization and interaction of the HCO₃⁻ transporters at the resting (left) and stimulated states (right). E-Syt3 set the epithelial resting secretory state by keeping PtdSer at the basolateral and luminal ER/PM junctions at a lower level. At the E-Syt3 LM domain low PtdSer the cAMP signaling pathway is not assembled and access of the PKAc to CFTR is prevented. At the BLM E-Syt3 domain, low PtdSer restricts the access of NBCe1-B to IRBIT. Recruitment of ORP5 to dissociate between the transporters and E-Syt3 and an increase in junctional PtdSer restore LM cAMP signaling to activate CFTR and BLM co-localization of NBCe1-B and IRBIT and initiation of the epithelial fluid and HCO₃⁻ secretory state. Source data are available online for this figure.

(Bevers and Williamson, 2016; Muallem et al, 2017). The level of PtdSer is variable in various cellular membranes, with high levels at the plasma and lysosomal membranes (Bevers and Williamson, 2016; Kay and Fairn, 2019). The junctional level of PtdSer is determined by lipid transfer proteins (LTP) like the PI(4)P/PtdSer exchangers ORP5 and ORP8 (Chung et al, 2015; Chung et al, 2023;

Sohn et al, 2018), and likely by other LTPs. In the present work, we show that junctional PtdSer is also regulated by the E-Syts. The regulation by E-Syts is mediated by their SMP domains that can act acutely and chronically to determine the steady-state level of junctional PtdSer. Acute targeting of the SMP domain to FRB at the caveola was sufficient to reproduce the effect of E-Syt3. Moreover,

the role of the E-Syts on junctional PtdSer is exquisitely specific. Measurement of PtdSer by TIRF that reports mainly PtdSer at the junctional subdomains indicated that E-Syt2 and E-Syt3 similarly reduced junctional PtdSer, yet when present together the effect of E-Syt3 dominated over the effect of E-Syt2 in regulating CFTR and E-Syt2 does not appear to regulate NBCe1-B activity. The specificity is dictated by localization of the SMP domain by the third E-Syts C2 domains. All E-Syts have PM targeting C2 domains (Giordano et al, 2013; Sclip et al, 2016) but only E3C2C targets the SMP domains of all E-Syts to the transporters to inhibit transporters activity, while targeting the SMP domains by E2C2C modestly activated the transporters (Fig. 2B,C). Together, these findings imply the existence of multiple distinct junctional PtdSer nanodomains accessed by each of the E-Syts to regulate specific cellular functions. In this respect, E-Syt1, but not E-Syt2 and E-Syt3, regulates the activation of Orai1 by STIM1 (Maleth et al, 2014). E-Syt2, but not E-Syt1 and E-Syt3, position the lipid phosphatase Sac1 at the ER/PM junctions (Dickson et al, 2016) and we show here that E-Syt3, but not E-Syt2 and E-Syt1 regulate CFTR activity and NBCe1-B activity.

An opposite effect of E-Syt3 on the transporters was observed with ORP5, while ORP8 had no significant effects (Figs. 4 and 7). ORP5 and ORP8 reciprocally affect the ER/PM junctional PtdSer, which is markedly increased by ORP5 and slightly reduced by ORP8 ((Chung et al, 2023) and Appendix Fig. S12). As with the E-Syts, the reciprocal effects of the ORPs are also due to differential localization at the ER/PM junctions that is specified by their PH domains (PHD). Localization by their PHD determines their function, which is reversed when their PHDs are switched (Chung et al, 2023). ORP5 reversed the inhibitory effect of E-Syt3 that required PtdSer transport by ORP5. Expression of ORP5 did not activate the transporters, likely because the native ORP5 was sufficient for full activation. This possibility is supported by the reduction in PtdSer and transporter activity caused by depletion of the native ORP5. The reduced PtdSer level and transporters activity by depletion of the native ORP5 was reversed by concomitant depletion of E-Syt3 (Fig. 4E). Moreover, ORP5 dissociated the interaction between CFTR and NBCe1-B with their inhibitor E-Syt3 and increased their interaction with their activators, the cAMP pathway and IRBIT, respectively (Fig. 7). Together, these findings suggest that E-Syt3 and ORP5 access the same ER/PM junctional PtdSer pool to reciprocally stabilize the transporters inactive state by E-Syt3 and the active state by ORP5, as depicted in the model in Fig. 9F.

Lipids affect membrane protein activity by several mechanisms. Lipids determine the membrane hydrophobicity, thickness, and curvature, all of which affect the structure and thus the activity of lipid-embedded proteins (Levental and Lyman, 2023; Renard and Byrne, 2021). Lipids can also bind to specific domains of the proteins to directly modulate their activity (Dickson and Hille, 2019; Renard and Byrne, 2021). Limited evidence suggests that lipids can control the assembly of protein complexes, with the best evidence for the role of cardiolipin in organization of the mitochondrial electron transport chain (Paradies et al, 2019). We describe here a new mode of regulation by lipids, specifically by PtdSer at the ER/PM junction that is controlled by E-Syt3. The junctional PtdSer modulates assembly of signaling complexes and their signal transduction. Multiple evidence indicate that PtdSer does not appreciably affect CFTR and NBCe1-B conductance.

When the transporters are in an active state, whether set by pre-activation with agonists or by mutations, they are resistant to changes in PM PtdSer. Rather, PtdSer is required for assembly of the cAMP/PKA signaling complex and activation of CFTR by the cAMP pathway. PtdSer is also required for activation of NBCe1-B by IRBIT (Fig. 7). Hence, PtdSer controls the access of the transporters to their signaling pathways at the junctions. This likely accounts for the findings that the constitutively active transporters do not prevent the Co-IP and FRET between CFTR and E-Syt3 and NBCe1-B and IRBIT.

Regulation by PtdSer requires localization of signaling pathways in specific PtdSer domains and PtdSer sensing by CFTR and NBCe1-B. Clustering of PtdSer in plasma membrane nanodomains have been reported in several studies (Hirama et al, 2017; Kefauver et al, 2024; Lenoir et al, 2021; Zhou and Hancock, 2018, 2023). We identified PtdSer sensors in CFTR and NBCe1-B that are required for their regulation by PtdSer. The most likely CFTR PtdSer sensor is the lasso domain, which preferentially binds PtdSer over PtdC and PI(4)P, and mutation of the lasso domain positive charges eliminates regulation by E-Syt3 and by PtdSer hydrolysis and prevents binding of PtdSer to the recombinant lasso domain (Fig. 6). Previous studies reported that PtdSer increases CFTR NBD1 ATPase activity (Hildebrandt et al, 2017). This PtdSer is unlikely to be the CFTR PtdSer sensor affected by E-Syt3 since NBD1 location is quite far from the PM (Liu et al, 2017; Zhang and Chen, 2016; Zhang et al, 2017), and mutation of NBD1 surface positively charged residues failed to prevent regulation of CFTR by E-Syt3 (Appendix Fig. S16B). The PtdSer sensor of NBCe1-B is the N-terminus autoinhibitory domain (AID) that binds PtdSer over PtdC, deletion of the AID prevents regulation of NBCe1-B by E-Syt3 and by PtdSer hydrolysis and mutation of the positively charged R42/R43/R44 eliminated regulation by PtdSer (Figs. 4 and 5). Previously we reported that mutation of the same arginine residues prevented interaction of NBCe1-B with IRBIT and its stimulation by $PI(4,5)P_2$, although it was necessary to mutate all 3 arginines to see these effects (Hong et al, 2013), while it was sufficient to mutate R44 to eliminate binding of PtdSer to the AID and regulation of NBCe1-B by E-Syt3 (Fig. 5C,H). This suggests that binding PtdSer to the AID is more sensitive than binding $PI(4,5)P_2$. Moreover, it is possible that PtdSer and $PI(4,5)P_2$ compete for interaction with the same site to regulate NBCe1-B activity. Since CFTR is also regulated by $PI(4,5)P_2$ (Cottrill et al, 2020; Himmel and Nagel, 2004), it is possible that also PtdSer and $PI(4,5)P_2$ compete for regulation of CFTR by interacting with the lasso domain.

The signaling and functional role of E-Syt3 in epithelial fluid and electrolyte secretion is illustrated in the model in Fig. 9F. The main findings of the current studies are that at rest E-Syt3 maintains low PtdSer at the selective luminal and basolateral junctions to prevent activation of CFTR by the cAMP pathway and of NBCe1-B by IRBIT. The effects of E-Syt3 are reversed by ORP5 that increases the level of PtdSer at the same ER/PM junction. PtdSer is required for activation of both transporters. Therefore, to initiate the active state the activity of E-Syt3 is inhibited by a mechanism that recruits ORP5 to the junction, resulting in increased junctional PtdSer that facilitates activation of CFTR by the cAMP/PKA pathway and of NBCe1-B by IRBIT. Targeted transgenic overexpression and siRNA depletion of E-Syt3 in the mouse salivary ducts revealed a significant role for E-Syt3 in ductal

 

electrolyte secretion. In addition, depletion of E-Syt3 increased pancreatic duct $Cl^-$ fluxes by CFTR and increased forskolin-stimulated ductal fluid secretion. Depletion of native E-Syt3 had no effect on saliva fluid volume but reduced saliva $Na^+$ and $Cl^-$ similar to knockout of CFTR (Catalan et al, 2010). NBCe1-B and CFTR have similar functions in the intestine (Kunzelmann et al, 2017; Seidler, 2024), lung (Quinton, 2010), and bile duct (Jung and Lee, 2014), making our finding relevant to all epithelia expressing CFTR. Together, our findings allow us to suggest that E-Syt3 set the epithelial resting secretory state by reducing PtdSer at selective basolateral and luminal junctional domains where CFTR and NBCe1-B are located. Reduction of the LM PtdSer by E-Syt3 disrupts the cAMP signaling pathway and the access of the PKAc to CFTR (and likely other cAMP-regulated transporters and proteins). Reduction in BLM PtdSer restricts the access of NBCe1-B to IRBIT. Recruiting ORP5 to the junctions dissociates E-Syt3 from the transporters and increases junctional PtdSer to restore LM cAMP signaling to activate CFTR and BLM co-localization of NBCe1-B and IRBIT and initiation of the epithelial fluid and $HCO_3^-$ secretory state.

Our findings have broad implications to many diseases of $HCO_3^-$ secretion, including CF, pancreatitis, Sjögren's disease, diarrhea and cholestatic liver disease. In all these diseases $HCO_3^-$ secretion is markedly reduced and can be attributed to altered CFTR activity independent of mutations in CFTR. Persistent inhibition of CFTR or NBCe1-B by E-Syt3 may contribute further to reduced $HCO_3^-$ secretion and progression of the diseases. To further understand the role of E-Syt3 in $HCO_3^-$ secretion and in the diseases, it is necessary to determine how PtdSer transport by E-Syt3 is regulated in the transition from the resting to the stimulated secretory state.

# Methods

**Reagents and tools table**

| Reagent/resource | Reference or source | Identifier or catalog number |
|---|---|---|
| **Experimental models** | | |
| Cell line- HEK293T | ATCC | |
| Mouse Line C57BL/6 | Jackson Lab | |
| **Recombinant DNA** | | |
| E-Syt1 | Dr. Pietro De Camilli (Yale University) | |
| E-Syt2 | Dr. Pietro De Camilli (Yale University) | |
| E-Syt3 | Dr. Pietro De Camilli (Yale University) | |
| ORP5 | Dr. Pietro De Camilli (Yale University) | |
| ORP8 | Dr. Pietro De Camilli (Yale University) | |
| AKAP79-mTurqoise2 | Dr. Mark Dell'Acqua (University of Colorado) | |
| GFP-evt2-2XPH | Dr. Tamas Balla (NICHD/NIH) | |

| Reagent/resource | Reference or source | Identifier or catalog number |
|---|---|---|
| NES-EGFP-P4Mx1 | Addgene | Plasmid #108121 |
| PH-PLCD1-GFP | Addgene | Plasmid #51407 |
| **Antibodies** | | |
| Rabbit anti-GFP | Invitrogen | #2273763 |
| Mouse anti-Myc | Cell Signaling technology | #2276 |
| Mouse anti-RFP | Rockland | # 200-301-379 |
| Mouse anti-Flag | Millipore Sigma | #B3111 |
| Mouse anti-HA | Cell Signaling technology | #2367 |
| Mouse anti-AKAP79 | Santa Cruz Biotechnology | # Sc-17772 |
| Mouse anti-PKAc | Biorad | #VMA 00679 |
| Mouse anti-CFTR | Millipore | #05-583 |
| Rabbit anti-E-Syt3 | Bioss | #bs-12165R |
| Rabbit anti-GST | Invitrogen | #A5800 |
| Rabbit anti-6xHN | Takara | #631213 |
| **Oligonucleotides and other sequence-based reagents** | | |
| PCR Primers | This study | Supplementary Table S1 |
| qPCR Primer for human E-SYT1 | Applied Biosystems | Hs00248693-m1 |
| qPCR Primer for mouse E-SYT1 | Applied Biosystems | Mm00803645_m1 |
| qPCR Primer for human E-SYT2 | Applied Biosystems | Hs00393482-m1 |
| qPCR Primer for mouse E-SYT2 | Applied Biosystems | Mm00466339_m1 |
| qPCR Primer for human E-SYT3 | Applied Biosystems | Hs00736518-m1 |
| qPCR Primer for mouse E-SYT3 | Applied Biosystems | Mm00625223_m1 |
| qPCR Primer for human GAPDH | Applied Biosystems | Hs 02786624-g1 |
| qPCR Primer for mouse GAPDH | Applied Biosystems | Mm99999915_g1 |
| **Chemicals, enzymes, and other reagents** | | |
| Lipofectamine 2000 | Life technologies | 11668019 |
| Fluoromount-G (Electron Microscopy Sciences, Hatfield, PA, USA) | | |
| Annexin V-FITC | Invitrogen | #A13199 |
| Isopropyl β-D-1-thiogalactopyranoside (IPTG) | Invitrogen | #15529-019 |
| B-PER® Bacterial Protein Extraction Reagent | Thermo Fisher Scientific | #90084 |
| TRIZOL | Invitrogen | 15596026 |
| cDNA reverse transcription kit | Applied Biosystems | 4368814 |
| Annexin V-FITC | Invitrogen | #A13199 |
| HisTalon gravity columns | Takara | #635654 |
| Pierce GST Spin Purification Kit | Thermo Scientific | # 16107 |
| Phosphotidyl Serine (PtdSer) Beads | Echelon Biosciences | P-BOPS |
| Phosphotidyl Choline (PtdC) Beads | Echelon Biosciences | P-BOPC |
| PI(4,5)P2 Beads | Echelon Biosciences | P-B045a |

| Reagent/resource | Reference or source | Identifier or catalog number |
|---|---|---|
| Brain PtdSer | Avanti Polar Lipids | Cat # 6505--6505 |
| Brain PtdC | Avanti Polar Lipids | Cat # 840053C-25mg |
| Brain PtdE | Avanti Polar Lipids | Cat # 840022 |
| VQAd MYC-E-Syt3 serotype 5 adenovirus | ViraQuest Inc. | |
| Chloride Assay kit | Abnova | KA1645 |
| **Software** | | |
| GraphPad Prism 10 | | |
| pClamp 11 software | (Molecular Devices) | |
| GraphPad 9 | | |
| FluoView software (FV10-ASW 4.2) | Olympus | |
| ImageJ | | |
| NIS-Elements of (Nikon) | Nikon | |
| Zen Black | Carl Zeiss | |
| Zen Blue | Carl Zeiss | |
| **Other** | | |

## Constructs, cell transfection, siRNA treatment

Appendix Table S1 lists the primers used to generate all constructs used at the present work. CFTR and several mutants are described in (Shcheynikov et al, 2015) and NBCe1-B and mutants in (Hong et al, 2013). Several plasmids were generously provided by colleagues: E-Syt1, E-Syt2, E-Syt3, ORP5 and ORP8 are described in (Chung et al, 2015; Giordano et al, 2013) and EVT2-XPH in (Chung et al, 2023). Mutants and deletions in E-Syt1, E-Syt2, E-Syt3, NBCe1-B and CFTR were generated using the QuikChangeTM Lightning Site-Directed Mutagenesis kit (Santa Clara, CA, USA). All constructs were verified by sequencing the entire ORFs. The following constructs were generated: E-Syt3(R853/R854/K855A), E-Syt3(K123/I124A), E-Syt3(K174S), E-Syt3ΔC2C (1–754), E3C2C (754–886), E-Syt3ΔSMP (Δ114–291), E-Syt3(E2SMP), E-Syt2(E3SMP), E1SMP (1–313), E2SMP (1–370), E3SMP (1–293), E2C2C(786–921), E3SMP(114–293), E3SMP(K207EI/208E), CFTR(R25/K26A),CFTR(K64A/K65A), CFTR(R74A/R75A), CFTR(R774A/R775A/R776A), CFTR(Q1042A/E1046A), CFTR(S1049A/F1052A), CFTR (S1155/R1158A), CFTR (K612A/K613A/K615 A), CFTR (R29/R31A), CFTR(K52/R55/R59) and CFTR(K64/K65/K68A) and NBCe1-B His-1-95. The PLA1a1, generation of PLA1a1-FKBP and targeting the PLA1a1 to the PM with the PFBP-FRB system have been described before (Chung et al, 2023).

HEK293T cells were grown at 37 °C with 5% $CO_2$ in a DMEM media supplemented with 10% FBS. Cells were plated in six-well plates and grown to 50–60% confluency, and were transfected 16–18 h before recording. Transfection was with Lipofectamine 2000 (catalog 11668019; Life Technologies) according to the manufacturer's instructions. For silencing of proteins, cells were treated with scrambled siRNA or with siRNA targeted against the desired protein. siRNAs were selected after testing three separate probes and selecting the most effective for experiments. After 48 h, the cells were transfected with plasmids for expression of the desired proteins and were used for molecular and protein analysis or current recording 24 h later.

## Quantitative PCR analysis

Total RNA was isolated from HEK293T cells and mouse submandibular glands using TRIZOL (Invitrogen) according to the manufacturer's protocol. Total RNA (2 µg) was used to synthesize cDNA using high-capacity cDNA reverse transcription kit (Applied Biosysytems) with human and mice primers for quantitative reverse-transcriptase PCR for E-SYT1 (Hs00248693-m1) and (Mm00803645_m1), E-SYT2 (Hs00393482-m1) and (Mm00466339_m1), E-SYT3 (Hs00736518-m1) and (Mm00625223_m1) and GAPDH (Hs 02786624-g1) and (Mm99999915_g1) purchased from Applied Biosystems (Foster City, CA). The fold change in transcript levels was calculated by normalizing the threshold values to GAPDH.

## Preparation of SMG acini and duct suspension

All procedures for maintaining the mice and for isolation of acini and ducts followed NIH guidelines and were approved by the Animal Care and Use Committee of NIDCR. Freshly isolated mixture of acini and ducts were prepared by collagenase digestion as detailed before (Luo et al, 2001). In brief, SMGs were dissected out of mice treated with control virus, Adv-myc-Esyt3 infected, scrambled or siEsyt3 infused mice by limited collagenase digestion as detailed previously (Luo et al, 2001). After isolation, the SMG were washed and re-suspended in solution A containing (140 mM NaCl, 5 mM KCl, 10 mM glucose, 10 mM HEPES (pH 7.4 with NaOH), 1 mM $MgCl_2$, 1 mM $CaCl_2$), 0.02% soybean-trypsin inhibitor and 0.1% bovine serum albumin and kept on ice until use. To prepare ductal cells, the SMG was finely minced, and the minced SMG glands were incubated in solution A containing 2.5 mg/10 ml collagenase P for 5 min at 37 °C. The digest was washed with PBS and treated for 2 min at 37 °C with 0.05% trypsin-EDTA solution, washed with solution A, and re-treated with the solution containing collagenase for 3–4 min at 37 °C. Finally, the cells were washed with solution A and kept on ice until use.

## Microdissection and culture on pancreatic intralobular ducts

Pancreatic ductal fragments were isolated as described earlier (Maleth et al, 2015; Park et al, 2013). Briefly, terminal anesthesia with pentobarbital was followed by surgical removal of the pancreas. Following physical fragmentation, pancreatic tissue was digested in a vertical shaker with an enzymatic solution containing 100 U/ml collagenase, 0.1 mg/ml trypsin inhibitor, 1 mg/ml bovine serum albumin (BSA) in DMEM/F12 for 30 min at 37 °C. Small interlobular ducts were identified and isolated by microdissection under a stereomicroscope. The ducts were cultured in DMEM supplemented with 10% fetal bovine serum at 37 °C on filter membranes and treated with scrambled or the desired siRNA for 48 h before use. Ducts were transfected within 1 h after dissection. The siRNA was diluted in 250 µl Opti-MEM I, and 5 µl Lipofectamine 2000. After 24 h, the medium was replaced with a fresh medium without the siRNA, and if necessary, the ducts were cut into half fragments to release the accumulated fluid and tension.

The resealed ducts were used 48 h after the beginning of the transfection to measure fluid secretion.

## Measurement of fluid secretion by the sealed ducts

Fluid secretion was measured by video microscopy as described previously (Hong et al, 2015; Park et al, 2013). The sealed ducts were transferred to a perfusion chamber and perfused with HEPES- and then $HCO_3^-$-buffered media and stimulated with 1 μM forskolin. Images were captured at 2 min intervals obtained up to 40 min and analyzed offline by calculating the lumen volume as the ratio $V_t/V_0$, which was calculated using the equation $V_t/V_0 = (A_t/A_0)^{3/2}$, with A as the duct area. The standard HEPES-based solution contained (mM) 140 mM NaCl, 5 mM KCl, 1 mM $MgCl_2$, 1 mM $CaCl_2$, 10 mM HEPES (pH 7.4 with NaOH), and 10 mM glucose. The $HCO_3^-$-buffered solution was prepared by replacing 25 mM NaCl with 25 mM $Na^+$-$HCO_3^-$ and reducing HEPES to 2.5 mM. $HCO_3^-$-buffered solutions were gassed with 5% $CO_2$ and 95% $O_2$.

## Measurement of ductal $Cl^-$ fluxes with MQAE

Measurement of ductal $Cl^-$ with MQAE was as described before (Molnar et al, 2020). In brief, pancreatic ducts treated with control siRNA or siE-Syt3 were incubated in bath solution containing 2 mM MQAE for 30 min at 37 °C. The ducts were transferred to a perfusion chamber mounted on an Olympus IX71 inverted microscope and imaged with an Olympus MT-20 illumination system equipped with a 150 W xenon arc light source. MQAE was excited at 340 nm and light emitted at 510/84 nm was corrected, while the ducts were continually perfused with worm (37 °C) $HCO_3^-$-buffered solution described above or a $Cl^-$-free solution in which all $Cl^-$ was replaced with $NO_3^-$. The fluorescent signal was captured by a Hamamatsu ORCA-ER CCD camera using X20 oil immersion objective (Olympus; NA: 0.8) with a temporal resolution of 1 ses. Image analysis was performed by Olympus excellence software and results transferred to Prism software for illustration.

## Confocal microscopy and staining with antibodies and Annexin IV

For confocal imaging, HEK293T expressing tagged constructs were grown on glass coverslips, washed twice with PBS, and fixed by incubation with cold methanol for 20 min at −20 °C or with 4% paraformaldehyde and permeabilized by incubation with 0.2% Triton X-100 and 0.3% Saponin at room temperature for 10 min. After fixation, non-specific sites were blocked with 5% goat serum. Cells were stained with primary antibody against proteins of interest overnight and then for 1 h at room temperature with fluorescent secondary antibodies. Coverslips were mounted on glass slides with Fluoromount-G (Electron Microscopy Sciences, Hatfield, PA, USA) and analyzed. To detect externalized PtdSer HEK293T cells in 35 mm glass-bottom dishes were incubated with standard media (in mM: 140 NaCl, 5 KCl, 10 Hepes- pH 7.4, 10 glucose 1 $MgCl_2$, and 3 $CaCl_2$) containing, 1:25 dilution of annexin V-FITC (#A13199, Invitrogen) and imaged before and after treatment with 25 μM CPA for 15 min at room temperature. The images were recorded with using an Olympus confocal microscope (FV1200) equipped with UplanSApo X 40 oil immersion objective lens (NA 1.25;) at 1× zoom. and FluoView software (FV10-ASW 4.2) and were processed with ImageJ.

## TIRF microscopy

Images were recorded with a Nikon system that included NIS-Elements paired with the Nikon Eclipse Ti and Perfect Focus System and equipped with Nikon N-Storm, Andor iXon Ultra Camera with EMCCD Sensor, D-Eclipse C1. The objective used is 60× TIRF objective lens (Nikon), 1.45 NA Oil immersion, infinity/0.10–0.22 DIC H, as described before (Liu et al, 2022). All experiments were at 37 °C and solution changes were made by perfusion. The size and intensity of the puncta were analyzed by NIS-Elements of (Nikon) after subtraction of background and normalized to area size. CFTR-YFP and NBCe1-B-eGFP were used to analyze their expression. PtdSer was analyzed with the PtdSer sensor GFP-evt2-2xPH (Chung et al, 2015) that has a PH domain that binds PtdSer. The PI(4)P sensor GFP-P4P and the $PI(4,5)P_2$ sensor PLCδ-PHD-GFP were obtained from Addgene and specifically bind the respective lipids (Sohn et al, 2019). The results are shown as mean ± SEM, and statistical analysis and figures were made using GraphPad Prism 10.

## FRET measurements

HEK293T cells were plated at low confluence on 35 mm glass-bottom dishes (MatTek Corporation) and transfected with fluorescent-tagged constructs using Lipofectamine 2000 (Invitrogen) and incubated for 16–18 h. FRET imaging was performed at 37 °C using a confocal system (FV1200; Olympus) equipped with UplanSApo ×60 oil immersion objective (NA 1.35; Olympus) controlled by FluoView software (FV10-ASW 4.2). Images were acquired at 10-s intervals at a scan rate of 12.5 μs/pixel. To minimize photobleaching, low laser power (2–3%) was used. Images were acquired and FRET ratios were exported to GraphPad 9 for statistical analysis and presentation.

## Superresolution microscopy

AiryScan images were acquired using a multipurpose laser scanning AiryScan microscope (LSM880; Carl Zeiss) with 32 spatial detectors (GAsP-spectral) with 63X/1.4 Plan-Apochromat oil DIC supported with Zen Black software. The GFP and mCherry fluorescence were visualized with a 488, 561 nm laser, respectively. Images were analyzed, and line intensity profiles were generated using Zen Blue software.

## Western blots and co-immunoprecipitation (Co-IP)

Cells and mouse tissues were extracted with ice-cold lysis buffer (1×PBS containing 10 mM Na pyrophosphate, 50 mM NaF, 1 mM Na orthovanadate, 1% Triton X-100, and a protease inhibitor cocktail, pH 7.4) by incubation for 1 h at 4 °C. The lysates were collected and cleared by centrifugation for 15 min at 12,000 rpm. Samples (50–100 μl) were incubated with the desired antibodies overnight at 4 °C. Sepharose beads were added, and incubation continued for the last 4 h, after which the beads were collected and washed 3 times with lysis buffer by centrifugation. The proteins were released from the beads and were recovered by warming at 37 °C for 30 min in SDS sample buffer. Protein concentration in the extracts was measured by the Bradford method, and proteins were analyzed for the co-IP. CFTR-GFP and NBCe1-B-eGFP were

detected with anti-GFP antibodies (#2273763, Invitrogen), E-Syts-Myc with anti-Myc antibodies (#2276, Cell Signaling Technology), E3C2C-RFP with anti-RFP antibodies (#42807, Rockland), E3SMP-Flag with anti-flag antibodies (#B3111, Millipore Sigma), IRBIT-HA with anti-HA antibodies (#2367, Cell Signaling Technology) and native AKAP79 and PKAc with anti-AKAP79 (# Sc-17772, Santa Cruz Biotechnology) and anti-PKAc (#VMA 00679, Biorad) antibodies. Native CFTR and E-Syt3 were probed with (#05-583, Millipore) and (#bs-12165R, Bioss), respectively. ImageJ was used the quantify band intensities of the blots.

## Protein synthesis and purification

The 6xHis-tagged 1–105 residues of NBCe1-B fragment was generated by PCR and was cloned into pET22b vector with T7 promotor and 6 histidine residues at the BglII site. Following transformation in BL21(DE3) competent cells and induction with 1 mM isopropyl β-D-1-thiogalactopyranoside (IPTG, #15529-019, Invitrogen) overnight at 25 °C, the proteins were extracted with B-PER® Bacterial Protein Extraction Reagent (#90084, Thermo Fisher Scientific) and were purified with HisTalon gravity columns (#635654, Takara). Preparation of the GST-Lasso plasmid (residues 1–80 of CFTR) (Gee et al, 2010). Mutant 6-his NBCe1-B (1–105)-R44A and GST- Lasso K52A/R55A/R59A were generated using site-directed mutagenesis following expression and purification as described above. Purification of the proteins was verified by Coomassie Blue staining, as well as by immunoblotting with GST monoclonal antibody (#A5800, Invitrogen) or 6xHN polyclonal antibody (#631213, Takara).

## Protein pull-down assays with lipid-coated beads

Beads impregnated with phospholipids were obtained from Echelon Biosciences. In total, 25 µl beads with PtdSer (P-B0PS), PtdC (P-B0PC) and PI(4,5)P2 (P-B045a) beads were collected by centrifugation (1000× $g$ × 5 min at 4 °C) and equilibrated with equal volume of binding-wash buffer (10 mM HEPES pH 7.4, 150 mM NaCl, and 0.1% Igepal). Pull-downs were performed by incubating 0.5 µg purified GST-Lasso or GST (control, ab81793, abcam) proteins with the beads for 1 h at room temperature. The beads were then washed 5 times with binding buffer by incubation for 10 min and collection by centrifugation. The bound proteins were eluted by heating the beads with a Laemmli sample buffer (Bio-Rad) at 95 °C for 5 min, and detection by immunoblotting with anti-GST antibody. For lipid binding of the 6xHis-tagged 1–105 residues of NBCe1-B fragment, 0.05 µg protein were mixed with PtdSer or PtdC beads for pull-downs, and the eluted bound proteins were detected by immunoblotting with 6xHA polyclonal antibody.

## Current measurements

Currents were measured in cells transfected with CFTR-YFP or NBCe1-B-eGFP and were identified by their fluorescence. To minimize scatter, only cells with plasma membrane fluorescence and with similar fluorescence intensity were selected for current measurements. *CFTR Current:* CFTR current was recorded in HEK293T cells transfected with CFTR or mutants using Lipofectamine TM2000. The pipette and bath solutions contained Cl⁻ as the only permeable anion. The pipette solution contained 150 mM NMDG-Cl, 1 mM MgCl₂, 1 mM EGTA, 0.5 mM ATP, and 10 mM

HEPES at pH 7.3 using TRIS, osmolarity adjusted to 290 mOsm using NMDG-Cl. The bath solution contained 150 mM NMDG-Cl, 1 mM MgCl₂, 1 mM CaCl₂, 10 mM HEPES and 10 mM Glucose at pH 7.4 using TRIS, with osmolarity adjusted to 310 mOsm using NMDG-Cl. Lipids or vehicle were added to pipette solution after their preparation (see below). The pipettes had a resistance between 3 and 6 MΩ when filled with pipette solution. Before current recording, the junction potential of each pipette was offset to 0 using the Axopatch 200B amplifier. Seal resistance was always more than 2 GΩ. The cell capacitance was between 4 and 25 pf. The current was recorded by 400-ms rapid alteration of membrane potential from −100 to +100 mV every 2 s from a holding potential of 0 mV. The currents were filtered at 1 kHz and sampled at 10 kHz. The current recorded at +100 mV was used to calculate current density as picoampere/picofarad (pA/pF). Axopatch 200B patch-clamp amplifier, Digidata −1440A, and pClamp 11 software (Molecular Devices) were used for data acquisition and analysis.

### NBCe1-B current

NBCe1-B current was measured in HEK293T cells transfected with NBCe1-B or mutants at room temperature. Patch pipettes had a resistance of 3–6 MOhms when filled with intracellular solution. The recording procedure was as above for CFTR. The pipette solutions contained 135 mM N-Methyl-D-glucamine Glucose, 5 mM N-Methyl-D-glucamine chloride (NMDG-Cl), 1 mM MgATP, 2 mM BAPTA, (pH 7.3 with TRIS), with osmolarity adjusted to 290 mOsm using N-Methyl-D-glucamine Glucose (NMDG-Glucose)). The basal Hepes-buffered bath solution was as above, except that NMDG-Cl was replaced with NaCl. Current was inhibited by switching to HCO₃⁻-buffered solution. The HCO₃⁻-buffered solution was prepared by replacing 25 mM NaCl with equimolar amount of NaHCO3, and the solution was equilibrated with 5%CO₂/95% O₂. The current was recorded, and a holding potential of 0 mV followed by 400-ms rapid alteration of membrane potential from −100 to +100 mV every 2 s. The current recorded at +100 mV was used to calculate current density. The currents were filtered at 1 kHz and sampled at 10 kHz and analyzed as above.

## Lipids handling

The brain PtdSer (Cat # 6505--6505), PtdC (Cat # 840053C-25mg) and PtdE (Cat # 840022) were purchased from Avanti Polar Lipids, Birmingham, USA. The desired lipid concentration was prepared fresh on the day of the experiment by pipetting the appropriate volume of the lipid stock solution to a glass vial. The Chloroform was evaporated at RT for 1 min. Appropriate volume of standard internal pipette solution was added, and followed by sonication 5 × 5 s with 10 s breaks between the sonication steps to dissolve the lipids. The Lipid stock solution was covered with N₂ and stored at −20 °C until use.

## Adeno-Myc-E-Syt3 administration and siE-Syt3 infusion

VQAd MYC-E-Syt3 serotype 5 adenovirus with MYC and E-Syt3 was generated by ViraQuest Inc., North Liberty, IA. To evaluate the vector, HEK293T cells were infected with Adv-myc-E-Syt3 at an MOI of 500 for 24 h and cell lysate were prepared and used to evaluate E-Syt3 expression (Appendix Fig. S5L). Viral delivery and salivary secretion measurement in mice were approved by the NIDCR Animal Care and Use Committee. Viruses were delivered to 8-week-old C57 mice. The mice were anaesthetized with ketamine (60 mg kg⁻¹) and xylazine

(8 mg kg$^{-1}$) I.M. VQAd MYC-E-Syt3 and Adv-GFP (vector control) were delivered to submandibular at $2 \times 10^{10}$ particles per gland by retrograde ductal infusion as described earlier (Hong et al, 2015). The expression of E-Syt3-myc in the glands of three mice is shown in Appendix Fig. S5M. For siE-Syt3 and scrambled siRNA, a total of 200 pmol siRNA was diluted in 400 μl of Opti-MEM with 5 μl Lipofectamine 2000 were prepared and after 25 min the siRNA-lipofectamine complex was delivered to submandibular glands at a volume of 50 μl per gland by retrograde ductal infusion. During the cannulation, 0.5 mg kg$^{-1}$ of atropine was used I.M. to inhibit saliva secretion to increase transduction efficiency.

## Measurement of Cl$^-$, Na$^+$, and K$^+$ in saliva

Saliva was collected from mice stimulated with 0.5 mg/Kg pilocarpine. Prior to the experiment, the mice body weight was measured, and saliva was collected every 5 min over 20 min duration. Collected saliva samples were stored at −80 °C until further analysis. Saliva samples were centrifuged at $1500 \times g$ for 10 min to clear cell debris and diluted 20-fold in deionized water, and Cl$^-$ concentration in the salivary fluid was analyzed using the Chloride Assay kit according to the manufacturer's protocol (KA1645, Abnova). Na$^+$ and K$^+$ concentrations were analyzed by the ion analysis core at NIDDK using Inductively-Coupled-Plasma Optical Emission Spectrometry (ICP-OES) (Maldonado et al, 2022). Samples were diluted 1:500 with 5% nitric acid and in triplicate from the indicated number of mice to yield the mean concentrations.

## Statistics

All experiments were repeated at least three times, and all data are given as mean ± SEM. Distribution normality was verified for all dataset GraphPad using Prism 10. The individual dots in the columns indicate separate cells analyzed. Statistical significance was determined by means of Student's $t$ test with GraphPad Prism 9 software. $P$ values are listed in the Figures, and $P$ values smaller than 0.05 are considered statistically significant.

## Data availability

All data and materials generated as part of these studies in our lab are available on request and upon satisfying NIH rules.

The source data of this paper are collected in the following database record: biostudies:S-SCDT-10_1038-S44318-025-00470-9.

## Peer review information

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

## Acknowledgements

We thank Dr. Pietro De Camilli (Yale University) for generously providing the plasmids coding for all E-Syts, ORP5, and ORP8, Dr. Tamas Balla (NICHD/NIH) for GFP-Evt2-2XPH, Dr. Mark Dell'Acqua (University of Colorado) for AKAP79-mTurqoise2 and AKAP-EYFP, and the NIDCR imaging core for the help with image acquisition and analysis. This work was funded by an intramural NIH grant NIH/NIDCR DE000735-15.

## Author contributions

**Paramita Sarkar**: Data curation; Formal analysis; Validation; Investigation; Writing—review and editing. **Benjamin P Lüscher**: Conceptualization; Data curation; Formal analysis; Validation; Investigation; Writing—review and editing. **Zengyou Ye**: Data curation; Formal analysis; Investigation; Writing—review and editing. **Woo Young Chung**: Data curation; Formal analysis; Investigation; Writing—review and editing. **Ava Movahed Abtahi**: Data curation; Formal analysis; Validation; Investigation; Writing—review and editing. **Changyu Zheng**: Data curation; Formal analysis. **Min Goo Lee**: Resources; Supervision; Investigation. **Árpád Varga**: Data curation; Formal analysis. **Petra Pallagi**: Data curation; Formal analysis. **József Maléth**: Supervision; Validation; Investigation; Writing—review and editing. **Malini Ahuja**: Data curation; Formal analysis; Supervision; Validation; Writing—review and editing. **Shmuel Muallem**: Conceptualization; Formal analysis; Supervision; Funding acquisition; Validation; Writing—original draft; Writing—review and editing.

Source data underlying figure panels in this paper may have individual authorship assigned. Where available, figure panel/source data authorship is listed in the following database record: biostudies:S-SCDT-10_1038-S44318-025-00470-9.

## Disclosure and competing interests statement

The authors declare no competing interests.

