## [Peer Review File · The EMBO Journal]

Lipid transporters E-Syt3 and ORP5 regulate epithelial ion transport by controlling phosphatidylserine enrichment at ER/PM junctions

Paramita Sarkar, Benjamin Luscher, Zengyou Ye, Woo Young Chung, Ava Movahed Abtahi, Changyu Zheng, Min Goo Lee, Arpad Varga, Petra Pallagi, Jozsef Maleth, Malini Ahuja, and Shmuel Muallem

Corresponding author(s): Shmuel Muallem (shmuel.muallem@nih.gov)

Review Timeline:

Submission Date:	23rd Oct 24
Editorial Decision:	26th Nov 24
Revision Received:	16th Mar 25
Editorial Decision:	14th Apr 25
Revision Received:	22nd Apr 25
Accepted:	9th May 25

Editor: William Teale

Transaction Report:

Dear Shmuel,

Thank you again for the submission of your manuscript entitled "A plasma membrane phosphatidylserine subdomain specified by E-Syt3 mediates assembly and transduction by the cAMP pathway to regulate epithelial transport" and for your patience during the review process. We have now received the reports from the referees, which I copy below.

As you can see from their comments, while referee #1 was enthusiastic about the manuscript, several technical concerns were raised elsewhere that have the potential to detract from the strong conclusions you draw. These issues will require your attention before your manuscript can be published in The EMBO Journal. I suggest we schedule a Zoom meeting, once you have had the opportunity to digest the reports, to discuss the feasibility and breadth required of any follow-up experiments. Let me know when would be a time that would suit you.

Based on the overall interest expressed in the reports, however, I would like to formally invite you to address the comments of all referees in a revised version of the manuscript. I should add that it is The EMBO Journal policy to allow only a single major round of revision and that it is therefore important to resolve the main concerns at this stage. I believe the concerns of the referees are reasonable and addressable, but please contact me if you have any questions, need further input on the referee comments or if you anticipate any problems in addressing any of their points. Please, follow the instructions below when preparing your manuscript for resubmission.

I would also like to point out that as a matter of policy, competing manuscripts published during this period will not be taken into consideration in our assessment of the novelty presented by your study ("scooping" protection). We have extended this 'scooping protection policy' beyond the usual 3 month revision timeline to cover the period required for a full revision to address the essential experimental issues. Please contact me if you see a paper with related content published elsewhere to discuss the appropriate course of action.

Again, please contact me at any time during revision if you need any help or have further questions.

Thank you very much again for the opportunity to consider your work for publication. I look forward to your revision.

Best regards,

William

William Teale, Ph.D.
Editor
The EMBO Journal

When submitting your revised manuscript, please carefully review the instructions below and include the following items:

- 1) a .docx formatted version of the manuscript text (including legends for main figures, EV figures and tables). Please make sure that the changes are highlighted to be clearly visible.
- 2) individual production quality figure files as .eps, .tif, .jpg (one file per figure).
- 3) a .docx formatted letter INCLUDING the reviewers' reports and your detailed point-by-point response to their comments. As part of the EMBO Press transparent editorial process, the point-by-point response is part of the Review Process File (RPF), which will be published alongside your paper.
- 4) a complete author checklist, which you can download from our author guidelines ([https://wol-prod-cdn.literatumonline.com/pb-assets/embo-site/Author Checklist%20-%20EMBO%20J-1561436015657.xlsx](https://wol-prod-cdn.literatumonline.com/pb-assets/embo-site/Author%20Checklist%20-%20EMBO%20J-1561436015657.xlsx)). Please insert information in the checklist that is also reflected in the manuscript. The completed author checklist will also be part of the RPF.
- 5) Please note that all corresponding authors are required to supply an ORCID ID for their name upon submission of a revised manuscript.

6) We require a 'Data Availability' section after the Materials and Methods. Before submitting your revision, primary datasets produced in this study need to be deposited in an appropriate public database, and the accession numbers and database listed under 'Data Availability'. Please remember to provide a reviewer password if the datasets are not yet public (see <https://www.embopress.org/page/journal/14602075/authorguide#datadeposition>). If no data deposition in external databases is needed for this paper, please then state in this section: This study includes no data deposited in external repositories. Note that the Data Availability Section is restricted to new primary data that are part of this study.

Note - All links should resolve to a page where the data can be accessed.

8) For data quantification: please specify the name of the statistical test used to generate error bars and P values, the number (n) of independent experiments (specify technical or biological replicates) underlying each data point and the test used to calculate p-values in each figure legend. The figure legends should contain a basic description of n, P and the test applied. Graphs must include a description of the bars and the error bars (s.d., s.e.m.).

9) We would also encourage you to include the source data for figure panels that show essential data. Numerical data can be provided as individual .xls or .csv files (including a tab describing the data). For 'blots' or microscopy, uncropped images should be submitted (using a zip archive or a single pdf per main figure if multiple images need to be supplied for one panel). Additional information on source data and instruction on how to label the files are available at .

10) We replaced Supplementary Information with Expanded View (EV) Figures and Tables that are collapsible/expandable online (see examples in <https://www.embopress.org/doi/10.15252/embj.201695874>). A maximum of 5 EV Figures can be typeset. EV Figures should be cited as 'Figure EV1, Figure EV2" etc. in the text and their respective legends should be included in the main text after the legends of regular figures.

12) Our journal encourages inclusion of *data citations in the reference list* to directly cite datasets that were re-used and obtained from public databases. Data citations in the article text are distinct from normal bibliographical citations and should directly link to the database records from which the data can be accessed. In the main text, data citations are formatted as follows: "Data ref: Smith et al, 2001" or "Data ref: NCBI Sequence Read Archive PRJNA342805, 2017". In the Reference list, data citations must be labeled with "[DATASET]". A data reference must provide the database name, accession number/identifiers and a resolvable link to the landing page from which the data can be accessed at the end of the reference. Further instructions are available at .

13) In order to increase the reproducibility and reach of your work, The EMBO Journal includes a table of reagents that were used in the study. Please provide this along with your revisions.

We realize that it is difficult to revise to a specific deadline. In the interest of protecting the conceptual advance provided by the work, we recommend a revision within 3 months (24th Feb 2025). Please discuss the revision progress ahead of this time with the editor if you require more time to complete the revisions. Use the link below to submit your revision:

Referee #1:

General summary and opinion about the principal significance of the study, its questions and findings

This is an extremely interesting, topical, and novel research paper which I believe will have a significant impact not only in the field of epithelial ion and fluid transport, but as a new paradigm for regulation of protein function by PL transporters at the PM/ER junction.

The paper utilizes a very broad range of in vitro molecular/biochemical and fluorescent techniques that provide strong and conclusive evidence for a novel role of the PL transporter, E-Syt-3 in the acute regulation of two key ion transporters, CFTR and NBCe1-B, that control salt and fluid secretion in many epithelial tissues. The study also provides a detailed mechanistic understanding of how E-Syt-3 regulates these two transporters, building on previous work from the authors' labs. Importantly, the authors have also validated these findings in two different, physiologically relevant, epithelial tissues; salivary glands in vivo, and isolated, ex-vivo, pancreatic ducts. While the later studies are consistent with the in-vitro cell line work, there remains some concerns over the interpretation of the results and, therefore, a clear validation of the Ms major conclusions.

Specific major concerns essential to be addressed to support the conclusions.

1. Figure 7. Salivary gland work. Have the authors investigated co-localisation of E-Syt3 with either CFTR or NBCe1-B by immunofluorescence in native glands? Furthermore, it is not clear what role NBCe1-B plays in NaCl absorption in the salivary duct. This needs further discussion/explanation.

2. Figure 8. Pancreatic results. (i) Panels A/B. The effect of KD of Esyt-3 on 'Cl flux' is rather minimal and the summarised results show extensive scatter. Importantly, no evidence is provided that this occurs via CFTR. A more direct approach would be to perform patch clamp studies to address this crucial point. The fluid secretion studies look more robust, but again no evidence is provided that CFTR underlies the increase in fluid secretion. This is relevant because previous studies have shown that

mouse ducts also express the calcium-activated chloride channel which significantly contributes to salt and fluid secretion, in addition to CFTR (Pascua, et al., Pflugers Archive 2009). Furthermore, the authors have not shown that their siRNA approach significantly reduced E-syt3 protein levels in isolated ducts.

Minor concerns that should be addressed.

1. It is important to indicate the number of animals (N) as well as the number of technical replicates (n) used in figures 7 and 8.
2. Statistics. The authors have only employed parametric tests, but there is no information whether they checked if the results were normally distributed.

Additional, non-essential, suggestions for improving the study (which will be at the author's/editor's discretion).

The figures are very 'dense' and for a 'general' audience not easy to digest! Suggest some prioritization here to reduce the number of separate panels.

There are a number of typographical errors throughout the Ms, so careful editing is required.

Referee #2:

This manuscript presents a novel mechanism by which the lipid transfer protein E-Syt3 regulates the activity of two key epithelial ion transporters, CFTR and NBCe1-B, through modulation of PS levels at specialized membrane contact sites. The manuscript demonstrates that E-Syt3, but not the other E-Syts, selectively inhibits the activity of CFTR and NBCe1-B. This inhibition is mediated by the ability of the E-Syt3 C2C domain to target the SMP lipid transfer domain to PS nanodomains and decrease PS levels, which in turn impacts the assembly and function of signaling complexes required for activation of the transporters. The authors also identify the PS-sensing domains of the two transporters, providing evidence that increased PS caused by inhibiting E-Syt3 is required for activation of both transporters.

1. One limitation of the study is the reliance on overexpression experiments. The use of gene knockout (or knockdown) models would indeed provide stronger evidence for the biological relevance of the E-Syt3-PS axis. Assessing the impact of individual or combined E-Syt loss of function on the activity of CFTR and NBCe1-B would help elucidate potential compensatory mechanisms and increase the significance of these findings in a more physiologically relevant context. However, considering the compensatory and adaptive mechanisms of LTPs, depletion results might not be as promising as overexpression of E-Syt3, that brings the next limitation.
2. The exclusive focus on E-Syts as the target lipid transfer proteins (LTPs) is another limitation. While the E-Syts are certainly important players at ER-PM contact sites, it would be valuable to investigate whether the observed regulation of CFTR and NBCe1-B activity mediated by increased phosphatidylserine (PS) levels extends to other classes of LTPs present at membrane contact sites. Exploring the involvement of additional LTPs, at least in a preliminary manner, would strengthen the generalizability of the findings and provide a more comprehensive understanding of the lipid-based control mechanisms governing epithelial ion transport.
3. Regarding the binding affinity of the CFTR lasso domain (Fig 5F-H), the lack of a clear trend in the increasing protein concentration and the exact same size of GST and GST-Lasso are important aspects to address.
4. In addition, it is reported that CFTR is activated by PI(4,5)P₂, and C2C binds to PI(4,5)P₂. It would therefore be informative if the authors could show a direct comparison of the lasso domain's binding affinity for PS versus PI(4,5)P₂. Exploring the potential competition or cooperativity between these lipids in binding to the lasso domain and their impact on CFTR activity would further strengthen the mechanistic understanding of this regulation.
5. Regarding Fig 3A-B, the Evt2-2xPH biosensor is used to measure steady-state plasma membrane PS levels, but it can't specifically indicate the levels of PS at the ER-PM junctions. The authors should provide more direct evidence or references supporting the detection of junctional PS and PI(4,5)P₂ levels, as these spatial distributions are crucial for the proposed regulatory mechanism. Relatedly, in the beginning of one paragraph, it is written that "Since E-Syt3 prominently changes junctional PtdSer and PI(4,5)P₂", a reference should be provided for the detection of junctional PtdSer and PI(4,5)P₂ to support this statement.
6. In Fig 3H-K, the translocation of PLA1a1 to FRB at the PM induced large puncta formation, where PLA1a1 is colocalized with CFTR or NBCe1-B. Is this seen all the time?
7. The authors designed the ER-anchored SMP domains from the E-Syts, and PM caveolae-targeted SMP with the FRB/FKBP/rapamycin targeting system. Did the authors attempt to directly target the SMP to PM?
8. There are two versions of constitutively active CFTR used in different experiments. Clarifying the differences between them would help ensure the consistency and interpretability of the results.

9. Additionally, there were several mismatches between curves and quantitative column graphs, and the use of smaller sample sizes in some groups than others in the same experiment for certain comparisons. Ensuring a consistent and rigorous approach to sample size, data presentation, and statistical analysis is crucial for the reliability and reproducibility of the findings.

Referee #3:

Summary of the key results

In this study, Sarkar, Lüscher, and Ye et al. explored the potential role of lipid transfer proteins (LTPs), specifically E-Syts, which function at ER-PM contact sites, in regulating HCO₃⁻ transporters CFTR and NBCe1-B that mediate epithelial fluid and HCO₃⁻ secretion. They observed that overexpression of E-Syt3, but not E-Syt1 or E-Syt2, influences the currents of these transporters, potentially through its effect on "junctional PtdSer." Such property of E-Syt3 appears to require its SMP and C2C domains. While the majority of experiments relied on E-Syt3 overexpression in cultured cells, the authors also demonstrated that E-Syt3 knockdown affects salivary gland physiology and pancreatic ductal ion secretion.

Although some data presented in this study may suggest a potential role of PtdSer in regulating CFTR and NBCe1-B function, the evidence is insufficient to conclude that E-Syt3 specifically regulates PtdSer at PM nanodomains. Additionally, several key experiments lack proper controls or quantification. Overall, the study's conclusions are not fully supported by the presented data.

Major concerns:

- 1) While the manipulation of PtdSer appears to regulate CFTR and NBCe1-B currents, the data do not convincingly demonstrate that E-Syt3 regulates PtdSer levels at the junctional PM in a physiological context. First, all experiments assessing E-Syt3's impact on PtdSer levels were conducted using overexpression models. Furthermore, PtdSer levels were measured in an experiment using the biosensor evt2-2xPH, which may compete with E-Syt3's C2 domain for binding to the PM [Notably, E-Syt C2 domains interact with negatively charged lipids, including PI(4,5)P2 and PtdSer, in the inner leaflet of the PM bilayer (e.g., PMID: 26202220, PMID: 29222176)].
- 2) As the authors pointed out in the introduction, the SMP domain of E-Syts likely transports a wide variety of glycerolipids, including phospholipids and diacylglycerol (PMID: 27065097, PMID: 29222176). E-Syts form homo- and heteromeric complexes, and their lipid-transporting properties are highly controlled by cytosolic Ca²⁺ levels (PMID: 27065097, PMID: 29222176, PMID: 34916620). However, the study does not explain why PtdSer levels are specifically regulated by the SMP domain of E-Syts or how such regulation is controlled. Furthermore, the directionality of lipid transport in a physiological context (without overexpression) remains unaddressed.
- 3) The model depicted in Figure 8 is not well supported by the presented data. There is no evidence provided in this study to support that E-Syt2 and E-Syt3 act separately (rather than forming a hetero-meric complex) and maintain the segregation of PtdSer-depleted domain and PtdSer-rich domain (i.e., with endogenous levels of protein expression).
- 4) Co-expression of the E-Syt SMP domain with the C2C domain of E-Syt3 modulated CFTR and NBCe1-B currents (Figure 2), but this approach is not physiological, as these domains operate within the context of the full-length protein, making it difficult to interpret the data. Moreover, it is unclear how the C2C domain of E-Syt3 recruits the SMP domain of E-Syts to the PM when overexpressed, as shown in Figure 2C.
- 5) It is unclear why E-Syt3 co-immunoprecipitates with CFTR and NBCe1-B, as E-Syt3 is an ER-anchored protein and CFTR and NBCe1-B are presumably PM proteins. In the context of overexpression, a significant portion of CFTR and NBCe1-B may become trapped in the ER, leading to non-specific interactions with E-Syt3 in-cis. If the authors propose in-trans interactions, additional evidence is needed to support this claim.
- 6) The authors claim that the lasso domain of CFTR, located in the cytosolic leaflet of the PM bilayer, is crucial for sensing PtdSer to regulate CFTR function. If this is indeed the case, PtdSer exposure mediated by CPA should also affect CFTR function, as PtdSer levels in the inner leaflet would decrease upon exposure.
- 7) In the rapamycin-induced acute recruitment of proteins to the PM, why were the proteins targeted to the caveolar region? The method section does not clarify which constructs were used for this recruitment. Did the authors recruit the SMP domain to the PM without its ER-anchoring region? If so, how does this mediate lipid transport? It is essential to demonstrate whether this recruitment is robust and the acute recruitment indeed alters the lipid composition of the PM through appropriate control experiments.

8) It is unclear how the low levels of E-Syt3 were successfully detected using antibodies against E-Syt3. E-Syt3 is expressed at very low levels in both HEK cells and submandibular glands (SMG) (Extended View Figure 13). How was the specificity of the antibody validated for immunohistochemistry?

Other issues:

1. Introduction:

The first paragraph of the introduction could be refined for accuracy. For instance, there has not been strong evidence in the field suggesting that cholesterol is transported by the SMP domain of E-Syts. Some references cited pertain to yeast tricalbins, whereas the discussion focuses on mammalian E-Syt1/2. This paragraph should cite references more accurately and appropriately.

2. Experimental Controls and Quantification:

Some experiments lack proper controls and quantification. For example:

- Figures 1E-F do not convincingly support the claim that the surface expression of CFTR and NBCe1-B is unaffected by E-Syt3 overexpression. Could the overexpression of E-Syt3 trap these proteins in the ER? Might the artificial properties of E-Syt3 be regulated by its SMP and C2C domains?
- There is no evidence that the rapamycin-dependent recruitment of the E-Syt SMP domain or PLAA1 specifically affects PtdSer.
- Quantification is missing in Extended Figure 4.

3. Regulation of PtdSer:

The major regulator of PtdSer is ORP5/8 (PMID: 26206935). Can the authors assess the impact of manipulating these proteins on PtdSer levels?

4. PtdSer Regulation:

Global changes in PtdSer levels following the acute recruitment of PLAA1 appear sufficient to induce changes in CFTR and NBCe1 currents. This observation contradicts the proposed regulation of PtdSer at nanodomains.

5. Details of E-Syt Expression:

The results should clearly indicate whether E-Syts were overexpressed and, if so, specify the constructs used. Both Myc-tagged and FLAG-tagged proteins seem to have been utilized, but the methodology of the immunoprecipitation experiments is unclear.

6. Contradictory Findings:

There are several contradictory findings that make the data difficult to interpret. For instance, it is unclear why PtdSer hydrolysis following CFTR activation slightly increased CFTR activity (page 6).

7. Nanodomain Regulation by E-Syt3:

There is no evidence to support the claim that "the change in PtdSer is highly specific and localized in a subdomain accessed by E3C2C" (Page 6). With the current data, it remains unclear how E-Syt3 regulates PtdSer at the proposed nanodomains or whether such nanodomains even exist.

Response to reviewers' comments:

We greatly appreciate the positive evaluation of the manuscript and the constructive comments by all reviewers. In response to the comments, we conducted several additional experiments and expanded the scope of study, by exploring the role of ORP5 in regulating the transporters and antagonizing the effects of E-Syt3. The additional information was incorporated into the existing and new Figures and required an additional Figure to be added. To simplify the presentation, we transferred several panels to the Extended View section. The significant extensive edits are highlighted in yellow. The new Figures are listed below in the order of their appearance in the text.

1. Figures EV1B-C: Show effect of co-expressing E-Syt3+ E-Syt2 on CFTR and NBCe1-B currents to demonstrate the prominence of E-Syt3 over E-Syt2 in regulating CFTR activity.
2. Figures 1C and 1E: Show that depletion of the native E-Syt3 by siRNA increases CFTR and NBCe1-B currents.
3. Figures EV1H-I: Show no effect of depletion or expression of E-Syt3 on CFTR and NBCe1-B level at the TIRF field.
4. Figures EV5A-C: Provide TIRF images of the PtdSer, PI(4)P and PI(4,5)P₂ sensors used to measure the level of these lipids.
5. Figures 3A-C and D: Show effect of all E-Syts on PtdSer, PI(4)P and PI(4,5)P₂ levels (A-C) and of E-Syt3 on junctional PtdSer and PI(4)P.
6. Figures EV10C-D: We prepared the E3SMP(K207E/I208E)-FKBP mutant, in which the basic patches of the SMP domain that are used to interact with acidic lipids including PI(4)P and PI(4,5)P₂ were replaced. We show that the mutations eliminated inhibition of CFTR when E3SMP(K207E/I208E)-FKBP is expressed with E3C2C and when targeted to the FRB by rapamycin.
7. Figures EV12A-B: Show effect of ORP5, ORP8 and depletion of ORP5 and of ORP8 on junctional PtdSer and PI(4)P to show that ORP5 and E-Syt3 reciprocally regulate junctional PtdSer.
8. Figures EV12C-D: Show effect of siORP5, siE-Syt3 and siORP5+siE-Syt3 on junctional PtdSer and PI(4)P to show that native ORP5 and E-Syt3 likely access the same junctional PtdSer pool to antagonize the effect of each other on junctional PtdSer.
9. Figures EV12E-H: Show lack of effect of siORP5 (E-F) and ORP5 (G-H) on the level of CFTR and NBCe1-B in the TIRF field.
10. Figures 4D and 4I: Show effect of ORP5 and ORP5 lipid binding mutant on inhibition of CFTR (D) and NBCe1-B (I) activity by E-Syt3.
11. Figure 4E and 4J: Show effect of depletion of ORP5 and co-depletion of ORP5+E-Syt3 on CFTR (4E) and NBCe1-B (4J) currents to demonstrate that ORP5 and E-Syt3 reciprocally regulate transporters activity.
12. Figure 6G: Shows that the lasso domain poorly binds PI(4,5)P₂ relative to PtdSer.
13. Figures 7I, O: Show effect of ORP5 and of ORP8 on the interaction between CFTR (I) and NBCe1-B (O) with E-Syt3 by FRET.
14. Figures 7J-P: Show effects of ORP5 and ORP8 on CFTR-AKAP79 (J) and NBCe1-B-IRBIT (P) interaction by FRET.

15. Figures 9A and 9D: Show that pancreatic ductal Cl⁻ transport (A) and fluid secretion (D) are inhibited by the CFTR inhibitor CFTRInh₁₂₇.

16. Figure 9F: Revised the model to include the role of ORP5 in regulating epithelial fluid and HCO₃⁻ secretion.

Referee #1:

General summary and opinion about the principal significance of the study, its questions and findings

This is an extremely interesting, topical, and novel research paper which I believe will have a significant impact not only in the field of epithelial ion and fluid transport, but as a new paradigm for regulation of protein function by PL transporters at the PM/ER junction.

Response: Thank you very much for the strong positive comments. They are greatly appreciated.

The paper utilizes a very broad range of in vitro molecular/biochemical and fluorescent techniques that provide strong and conclusive evidence for a novel role of the PL transporter, E-Syt-3 in the acute regulation of two key ion transporters, CFTR and NBCe1-B, that control salt and fluid secretion in many epithelial tissues. The study also provides a detailed mechanistic understanding of how E-Syt-3 regulates these two transporters, building on previous work from the authors' labs. Importantly, the authors have also validated these findings in two different, physiologically relevant, epithelial tissues; salivary glands in vivo, and isolated, ex-vivo, pancreatic ducts. While the later studies are consistent with the in-vitro cell line work, there remains some concerns over the interpretation of the results and, therefore, a clear validation of the Ms major conclusions.

Response: We greatly appreciate the constructive comments. Based on those we have addressed them below with additional experiments, as suggested by the reviewers. We significantly expanded the scope of the studies by analyzing the role of ORP5 in regulating the transporters and in reversing the inhibitory effects of E-Syt3.

Specific major concerns essential to be addressed to support the conclusions.

1. Figure 7. Salivary gland work. Have the authors investigated co-localisation of E-Syt3 with either CFTR or NBCe1-B by immunofluorescence in native glands? Furthermore, it is not clear what role NBCe1-B plays in NaCl absorption in the salivary duct. This needs further discussion/explanation.

Response: Unfortunately, we are unable to perform colocalization due to suitable lack of antibodies such as rabbit/mice or rabbit/donkey for such experiments. However, we would like to point out the results in Figure 8A showing the Co-IP of the native ductal CFTR and E-Syt3. Co-IP is a much more stringent and rigorous test for interaction of the proteins than immunolocalization.

2. Figure 8. Pancreatic results. (i) Panels A/B. The effect of KD of E-syt-3 on 'Cl flux' is rather minimal and the summarised results show extensive scatter. Importantly, no evidence is provided that this occurs via CFTR. A more direct approach would be to perform patch clamp studies to address this crucial point. The fluid secretion studies look more robust, but again no evidence is provided that CFTR underlies the increase in fluid secretion. This is relevant because previous studies have shown that mouse ducts also express the calcium-activated chloride channel which significantly contributes to salt and fluid secretion, in addition to CFTR (Pascua, et al., Pflugers Archive 2009). Furthermore, the authors have not shown that their siRNA approach significantly reduced E-syt3 protein levels in isolated ducts.

Response: Thank you for raising this important point. We now show that the specific CFTR inhibitor CFTR-Inh₁₇₂ inhibits both the Cl⁻ flux measured with MQAE and ductal fluid secretion measured in the sealed duct.

Minor concerns that should be addressed.

1. It is important to indicate the number of animals (N) as well as the number of technical replicates (n) used in figures 7 and 8.

Response: As indicated in Figure 8 (original 7) panels E and K: 6 mice with vector, 8 mice with E-Syt3, 7 mice with scrambled siRNA and 7 mice with siE-Syt3. The information for the pancreatic ducts in Figure 9 is now included in the legend.

2. Statistics. The authors have only employed parametric tests, but there is no information whether they checked if the results were normally distributed.

Response: The statistics section states now "Distribution normality was verified for all data set using GraphPad Prism 10" that was done in response to your comment.

Additional, non-essential, suggestions for improving the study (which will be at the author's/editor's discretion).

The figures are very 'dense' and for a 'general' audience not easy to digest! Suggest some prioritization here to reduce the number of separate panels.

There are a number of typographical errors throughout the Ms, so careful editing is required.

Response: We appreciate the suggestion, and to improve clarity, we transferred substantial number of panels to the Extended View Figures and the simplified figures now show the essential information necessary to follow the main findings.

Referee #2:

This manuscript presents a novel mechanism by which the lipid transfer protein E-Syt3 regulates the activity of two key epithelial ion transporters, CFTR and NBCe1-B, through modulation of PS levels at specialized membrane contact sites. The manuscript demonstrates that E-Syt3, but not the other E-Syts, selectively inhibits the activity of CFTR and NBCe1-B. This inhibition is mediated by the ability of the E-Syt3 C2C domain to target the SMP lipid transfer domain to PS nanodomains and decrease PS levels, which in turn impacts the assembly and function of signaling complexes required for activation of the transporters. The authors also identify the PS-sensing domains of the two transporters, providing evidence that increased PS caused by inhibiting E-Syt3 is required for activation of both transporters.

1. One limitation of the study is the reliance on overexpression experiments. The use of gene knockout (or knockdown) models would indeed provide stronger evidence for the biological relevance of the E-Syt3-PS axis. Assessing the impact of individual or combined E-Syt loss of function on the activity of CFTR and NBCe1-B would help elucidate potential compensatory mechanisms and increase the significance of these findings in a more physiologically relevant context. However, considering the compensatory and adaptive mechanisms of LTPs, depletion results might not be as promising as overexpression of E-Syt3, that brings the next limitation.

Response: To address this concern, in addition to the findings with the mice (Figure 8) we measured the effect of E-Syt3 depletion by siRNA on CFTR and NBCe1-B activity. Moreover, in response to your comment 2, we included extensive studies with ORP5 and ORP8 and as part of these studies which

shows the effect of depleting ORP5 and ORP8 and effect of co-depletion of ORP5+E-Syt3 on junctional lipids and the function of CFTR and NBCe1-B.

2. The exclusive focus on E-Syts as the target lipid transfer proteins (LTPs) is another limitation. While the E-Syts are certainly important players at ER-PM contact sites, it would be valuable to investigate whether the observed regulation of CFTR and NBCe1-B activity mediated by increased phosphatidylserine (PS) levels extends to other classes of LTPs present at membrane contact sites. Exploring the involvement of additional LTPs, at least in a preliminary manner, would strengthen the generalizability of the findings and provide a more comprehensive understanding of the lipid-based control mechanisms governing epithelial ion transport.

Response: Thank you for this suggestion, which has helped us to significantly expand our studies. As part of addressing the comments we examined the effect of the other ER/PM junctional LTPs ORP5 and ORP8 that also regulate junctional PtdSer, we discovered that ORP5 serves to reverse the effects of E-Syt3. As a result, we included extensive studies on the effects of ORP5 and additionally the depletion of ORP5 on junctional lipids and the transporters (ORP8 and depletion of ORP8 had minimal or no significant effect on the transporters). The role of ORP5 is incorporated into the model and can be found in Figure 9F.

3. Regarding the binding affinity of the CFTR lasso domain (Fig 5F-H), the lack of a clear trend in the increasing protein concentration and the exact same size of GST and GST-Lasso are important aspects to address.

Response: Thank you for noticing this. This was our mistake of not noticing this, which occurred due to overexposing the blots. By reducing the background of the same blots as is done in the revised Figure clearly shows the effect of increased protein levels.

4. In addition, it is reported that CFTR is activated by PI(4,5)P₂, and C2C binds to PI(4,5)P₂. It would therefore be informative if the authors could show a direct comparison of the lasso domain's binding affinity for PS versus PI(4,5)P₂. Exploring the potential competition or cooperativity between these lipids in binding to the lasso domain and their impact on CFTR activity would further strengthen the mechanistic understanding of this regulation.

Response: We compared PtdSer and PI(4,5)P₂ binding to the lasso domain (Figures 6G and 6I). Importantly, please note that depletion of PI(4,5)P₂ had no effect on CFTR-E-Syt3 and NBCe1-B-E-Syt3 Co-IP, further indicating that PtdSer rather than PI(4,5)P₂ control the complexes (not current).

5. Regarding Fig 3A-B, the Evt2-2xPH biosensor is used to measure steady-state plasma membrane PS levels, but it can't specifically indicate the levels of PS at the ER-PM junctions. The authors should provide more direct evidence or references supporting the detection of junctional PS and PI(4,5)P₂ levels, as these spatial distributions are crucial for the proposed regulatory mechanism. Relatedly, in the beginning of one paragraph, it is written that "Since E-Syt3 prominently changes junctional PtdSer and PI(4,5)P₂", a reference should be provided for the detection of junctional PtdSer and PI(4,5)P₂ to support this statement.

Response: These sensors are very well established, most extensively by Tamas Balla's lab. We have now provided references for the development and use of the three sensors. In addition, Figure EV5A-C provides example images for all sensors at the TIRF field.

6. In Fig 3H-K, the translocation of PLA1a1 to FRB at the PM induced large puncta formation, where PLA1a1 is colocalized with CFTR or NBCe1-B. Is this seen all the time?

Response: Please see images of PLA1a1 translocation to plasma membrane and ER localized FRB in PMID: 37607230, which also shows the time course and extent of depletion of PM PtdSer by PLA1a1.

The images in Figure 4 show that after rapamycin treatment the PLA1a1 localize with CFTR and NBCe1-B and similar localization is observed in all experiments.

7. The authors designed the ER-anchored SMP domains from the E-Syts, and PM caveolae-targeted SMP with the FRB/FKBP/rapamycin targeting system. Did the authors attempt to directly target the SMP to PM?

Response: The SMP domain was targeted specifically to the ER/PM junctions with the FRB/FKBP system, and we have not used other targeting sequences. As indicated now, the SMP domain dimerized to form elongated lipid translocation pathway that can span the distance between the ER and PM. The dimer has basic patches at the two tip regions that bind to lipids (see Figure 3c in PMID: 36932127). When expressed at high concentration, the SMP dimer can transfer PM PtdSer (Figure EV8B) to regulate CFTR and NBCe1-B activity (Figure EV10A-B). The E3C2C domain facilitate transfer of the SMP domain to the junctions to mediate regulation of the transporters by lower SMP domain concentration (the combination of Figures EV3B, EV3D, 3H, 3I). To further demonstrate this, we mutated the SMP domain basic patches and show that the mutants inhibit the activity of the SMP domain (Figure EV10C-D).

8. There are two versions of constitutively active CFTR used in different experiments. Clarifying the differences between them would help ensure the consistency and interpretability of the results.

Response: There is no difference between them. The idea was to use more than one constitutively active CFTR mutant and show that irrespective of the constitutively active mutants, whether in NBD2 (E1371Q) or in the R domain (S1155RA/R1158A), the constitutively active mutants are not inhibited by depletion of PtdSer either by E-Syt3 or by PLA1a1. This is now indicated in the text.

9. Additionally, there were several mismatches between curves and quantitative column graphs, and the use of smaller sample sizes in some groups than others in the same experiment for certain comparisons. Ensuring a consistent and rigorous approach to sample size, data presentation, and statistical analysis is crucial for the reliability and reproducibility of the findings.

Response: We have made sure in all experiments the number of measurements meet statistical requirements.

Referee #3:

Summary of the key results

In this study, Sarkar, Lüscher, and Ye et al. explored the potential role of lipid transfer proteins (LTPs), specifically E-Syts, which function at ER-PM contact sites, in regulating HCO₃⁻ transporters CFTR and NBCe1-B that mediate epithelial fluid and HCO₃⁻ secretion. They observed that overexpression of E-Syt3, but not E-Syt1 or E-Syt2, influences the currents of these transporters, potentially through its effect on "junctional PtdSer." Such property of E-Syt3 appears to require its SMP and C2C domains. While the majority of experiments relied on E-Syt3 overexpression in cultured cells, the authors also demonstrated that E-Syt3 knockdown affects salivary gland physiology and pancreatic ductal ion secretion.

Although some data presented in this study may suggest a potential role of PtdSer in regulating CFTR and NBCe1-B function, the evidence is insufficient to conclude that E-Syt3 specifically regulates PtdSer at PM nanodomains. Additionally, several key experiments lack proper controls or quantification. Overall, the study's conclusions are not fully supported by the presented data.

Response: As detailed in the text, we have used several experimental paradigms showing that changes in PtdSer are key for the function of the channels: we show that E-Syt3 changes PtdSer; depletion of PtdSer by PLA1a1 reproduce the effect of E-Syt3; pharmacological reduction in PtdSer by two independent inhibitors of PtdSer synthetase inhibit the transporters like E-Syt3; most notably, PtdSer-but not PtdC, PtdEtn or PI(4,5)P₂- reverse the effects of E-Syt3 and pharmacological reduction in PtdSer. To even further increase the evidence in the role of PtdSer we now present analysis of the role of ORP5, another PtdSer transporter with opposite effect on junctional PtdSer than E-Syt3 and show that ORP5 that increases PtdSer antagonizes the effect of E-Syt3. Moreover, depletion of E-Syt3 and ORP5 have reciprocal effects on PtdSer relative to E-Syt3 and ORP5 and, accordingly, depletion of E-Syt3 increases the transporters activity, while depletion of ORP5 inhibits the transporters activity.

We believe that together the evidence can be used to conclude that E-Syt3 and ORP5 reciprocally regulate junctional PtdSer to affect activation of CFTR by the PKA pathway and NBCe1-B by IRBIT.

Major concerns:

1) While the manipulation of PtdSer appears to regulate CFTR and NBCe1-B currents, the data do not convincingly demonstrate that E-Syt3 regulates PtdSer levels at the junctional PM in a physiological context. First, all experiments assessing E-Syt3's impact on PtdSer levels were conducted using overexpression models. Furthermore, PtdSer levels were measured in an experiment using the biosensor evt2-2xPH, which may compete with E-Syt3's C2 domain for binding to the PM [Notably, E-Syt C2 domains interact with negatively charged lipids, including PI(4,5)P₂ and PtdSer, in the inner leaflet of the PM bilayer (e.g., PMID: 26202220, PMID: 29222176)].

Response: Obviously the sensor binds to PtdSer to detect PtdSer. However, the sensor does not deplete PtdSer. Please see Figure 6A in our recent publication in PMID: 37607230 showing that only after targeting PLA1a1 to the plasma membrane PtdSer is reduced and the sensor dissociates from the plasma membrane to the cytosol. Furthermore, the sensor is not present when the effects of E-Syt3 on CFTR and NBCe1-B activity are measured. Indeed E-Syt2 and E-Syt3 C3C domains binds to PI(4,5)P₂ and PI(4)P at the plasma membrane and mutation of the PI(4,5)P₂ binding site in E3C2C inhibits E-Syt3 effects. However, we show directly that beside functioning as E3C2C binding site, PI(4,5)P₂ does not affect regulation by E-Syt3 of signaling complexes formation by CFTR and NBCe1-B CFTR.

To extend the physiological role of NATIVE E-Syt3, and now native ORP5, we show the effect of depletion of E-Syt3 (and of ORP5 and both) by siRNA on PtdSer and PI(4)P and on CFTR and NBCe1-B activity. We also show that E-Syt3 and ORP5 reciprocally affect junctional PtdSer and transporters activity, likely by accessing the same junctional PtdSer pool.

2) As the authors pointed out in the introduction, the SMP domain of E-Syts likely transports a wide variety of glycerolipids, including phospholipids and diacylglycerol (PMID: 27065097, PMID: 29222176). E-Syts form homo- and heteromeric complexes, and their lipid-transporting properties are highly controlled by cytosolic Ca²⁺ levels (PMID: 27065097, PMID: 29222176, PMID: 34916620). However, the study does not explain why PtdSer levels are specifically regulated by the SMP domain of E-Syts or how such regulation is controlled. Furthermore, the directionality of lipid transport in a physiological context (without overexpression) remains unaddressed.

Response: First, please note that transfer of multiple glycerophospholipids was demonstrated only by the purified, reconstituted SMP domain. We are aware of only one study in live mammalian cells showing transfer of DAG and PtdSer, which is cited. Transfer of other lipids was measured. We show in Figure 3A-E that the E-Syts affect junctional PtdSer, PI(4)P and PI(4,5)P₂ levels in an isoform specific manner. Moreover, in yeast the tricalbins regulate PtdSer homeostasis (PMID: 35440494). Our justifications toward focusing on PtdSer were quite straightforward. First, we found that the effect of E-

Syt3 requires lipid transfer by the SMP domain, then we tested which lipid can reverse the effect of E-Syt3 including PA, PI(4,5)P₂, PtdC, PtdEtn and PtdSer reversed the effects of E-Syt3. Moreover, we then found that pharmacological and enzymatic depletion of PtdSer, but not PI(4,5)P₂, had the same effect as E-Syt3 on the function and assembly of complexes by CFTR and NBCe1-B. We now extended these quite compelling reasons in favor of PtdSer by showing that ORP5 that increases junctional PtdSer, antagonizes the effect of E-Syt3, while ORP8 that increases PI(4)P and PI(4,5)P₂ does not. Finally, we show that depletion of E-Syt3 and ORP5 by siRNA reciprocally affected junctional PtdSer and the function of the transporters. We believe that together these findings provide compelling evidence that PtdSer regulate the transporters and in addition show the role of the NATIVE E-Syt3, ORP5 and PtdSer.

3) The model depicted in Figure 8 is not well supported by the presented data. There is no evidence provided in this study to support that E-Syt2 and E-Syt3 act separately (rather than forming a heteromeric complex) and maintain the segregation of PtdSer-depleted domain and PtdSer-rich domain (i.e., with endogenous levels of protein expression).

Response: Thank you. We agree and revised the model to remove the E-Syt2 from the model. Considering the extensive new experiments with ORP5, we now include the role of ORP5 in the model.

4) Co-expression of the E-Syt SMP domain with the C2C domain of E-Syt3 modulated CFTR and NBCe1-B currents (Figure 2), but this approach is not physiological, as these domains operate within the context of the full-length protein, making it difficult to interpret the data. Moreover, it is unclear how the C2C domain of E-Syt3 recruits the SMP domain of E-Syts to the PM when overexpressed, as shown in Figure 2C.

Response: This is rather a confusing comment. Of course, the E-Syt3 physiology is mediated by the full-length E-Syt3. We asked which of the E-Syt3 domains are required for function. As detailed in response to comment 7 of reviewer 2, the SMP domain is assembled into dimers with positively charged residues that binds to lipids, the dimers span the space between the ER and PM to transfer lipids. The structure of E-Syt2 SMP domain with C2B and C2C show that the C2 domains interact with the SMP domain (PMID: 24847877). Therefore, we tested if the SMP domain and E3C2C are sufficient to mediate the function of E-Syt3 and it turned that they are. We think that these findings should be appreciated in identifying the critical and sufficient domains of E-Syt3 that mediate its function. Especially that we proceeded using these tools to show that the E3C2C can target any of the E-Syts SMP domains to the junctions to regulate the transporters, highlighting the incredible specificity of targeting by the C2C domains of the E-Syts that dictates their function.

5) It is unclear why E-Syt3 co-immunoprecipitates with CFTR and NBCe1-B, as E-Syt3 is an ER-anchored protein and CFTR and NBCe1-B are presumably PM proteins. In the context of overexpression, a significant portion of CFTR and NBCe1-B may become trapped in the ER, leading to non-specific interactions with E-Syt3 in-cis. If the authors propose in-trans interactions, additional evidence is needed to support this claim.

Response: This is an issue since E-Syt3 constitutively interacts with the PM and all proteins, E-Syt3, CFTR and NBCe1-B are localized at the junction as revealed by FRET and TIRF. The Co-IP is exactly the same as the PM localized Orai1 and the localized ER STIM1 that Co-IP when STIM1 binds to the PM by its polybasic domain. Moreover, we show that the NATIVE CFTR and E-Syt3 Co-IP! We show this by Co-IP, FRET and TIRF for the expressed proteins and by Co-IP of the NATIVE ductal proteins. These are all the available techniques to show interaction, and we utilized all for additional support.

6) The authors claim that the lasso domain of CFTR, located in the cytosolic leaflet of the PM bilayer, is crucial for sensing PtdSer to regulate CFTR function. If this is indeed the case, PtdSer exposure

mediated by CPA should also affect CFTR function, as PtdSer levels in the inner leaflet would decrease upon exposure.

Response: This is what we expected, and we show that this is not the case and explain that only TARGETED PtdSer depletion in the specific junction accessed by E3C2C. We show this in two ways: targeting the SMP domain and the PtdSer specific PLA1a1 to the junctions by **E3C2C** and the **FRB**/FKBP system works. Clearly, the general massive externalization of non-junctional PtdSer is not equivalent to junctional depletion of PtdSer.

7) In the rapamycin-induced acute recruitment of proteins to the PM, why were the proteins targeted to the caveolar region? The method section does not clarify which constructs were used for this recruitment. Did the authors recruit the SMP domain to the PM without its ER-anchoring region? If so, how does this mediate lipid transport? It is essential to demonstrate whether this recruitment is robust and the acute recruitment indeed alters the lipid composition of the PM through appropriate control experiments.

Response: The way the FRB/FKBP system works is that the FRB is palmitoylated that insert it in caveola that are within the junction. The FKBP domain is spliced with the protein to be targeted. Rapamycin binds to both proteins and attaches them together. The cited reference describes how the system works in further detail. The response to comment 7 of reviewer 2 and your comment 4 explains how the SMP domain without the ER anchor regulates PM PtdSer and the transporters. The new information if Figure EV10C-D shows that the positive charges at the tips of the SMP domain allow its function.

8) It is unclear how the low levels of E-Syt3 were successfully detected using antibodies against E-Syt3. E-Syt3 is expressed at very low levels in both HEK cells and submandibular glands (SMG) (Extended View Figure 13). How was the specificity of the antibody validated for immunohistochemistry?

Response: a) the E-Syt3 level is highly cell specific, and SMGs have high level. Nevertheless, evidently the level is high enough to regulate the function of the duct and the salivary glands; b) depletion of E-Syt3 in HEK cells and the ducts show that the level of the native E-Syt3 is sufficient to mediate its function; c) the E-Syt3 antibodies appear to be highly specific in western blots as they detected the correct band in transfected cells with E-Syt3. Moreover, another indication that the antibodies are highly specific for immunohistochemistry is that depletion of E-Syt3 nearly eliminated the staining in the duct.

Other issues:

1. Introduction:

The first paragraph of the introduction could be refined for accuracy. For instance, there has not been strong evidence in the field suggesting that cholesterol is transported by the SMP domain of E-Syts. Some references cited pertain to yeast tricalbins, whereas the discussion focuses on mammalian E-Syt1/2. This paragraph should cite references more accurately and appropriately.

Response: We have now cited a reference for PtdSer homeostasis by the tricalbins. As for cholesterol transfer by the SMP domain, please see SI Appendix, Fig. S5 in PMID: 27044075. We have also verified the citations are used correctly and appropriately.

2. Experimental Controls and Quantification:

Some experiments lack proper controls and quantification. For example:

- Figures 1E-F do not convincingly support the claim that the surface expression of CFTR and NBCe1-B is unaffected by E-Syt3 overexpression. Could the overexpression of E-Syt3 trap these proteins in the ER? Might the artificial properties of E-Syt3 be regulated by its SMP and C2C domains?

Response: We measured the effect of E-Syt3, siE-Syt3, (Figure EV1H-I) ORP5 and siORP5 (Figure EV12E-F) by TIRF field (the best resolution in live cells) to show no change in their level at the TIRF field.

- There is no evidence that the rapamycin-dependent recruitment of the E-Syt SMP domain or PLAA1 specifically affects PtdSer.

Response: For the SMP domain please see results in Figure EV8B, and in regard to PLA1a1 please see the time course and images in PMID: 37607230

- Quantification is missing in Extended Figure 4.

Response: (Now EV7). It is not possible to provide average when we see no PtdSer externalization. However, we now indicate in the Figure legend that the images represent results of 3 similar experiments.

3. Regulation of PtdSer:

The major regulator of PtdSer is ORP5/8 (PMID: 26206935). Can the authors assess the impact of manipulating these proteins on PtdSer levels?

Response: As described above and in the manuscript, we provide detailed studies on the role of ORP5 and ORP8 and their depletion (the first measurement on the effect of the depletion) on lipid levels and regulation of the transporters.

4. PtdSer Regulation:

Global changes in PtdSer levels following the acute recruitment of PLAA1 appear sufficient to induce changes in CFTR and NBCe1 currents. This observation contradicts the proposed regulation of PtdSer at nanodomains.

Response: This is not the case. It is important to note that PtdSer hydrolysis by PLA1a1 is not global, but it is targeted by the FRB/FKBP and specific at the junctions.

5. Details of E-Syt Expression:

The results should clearly indicate whether E-Syts were overexpressed and, if so, specify the constructs used. Both Myc-tagged and FLAG-tagged proteins seem to have been utilized, but the methodology of the immunoprecipitation experiments is unclear.

Response: Thank you. Please note that the original and current Methods section provide a complete list of the antibodies used, including the source and catalogue number.

6. Contradictory Findings:

There are several contradictory findings that make the data difficult to interpret. For instance, it is unclear why PtdSer hydrolysis following CFTR activation slightly increased CFTR activity (page 6).

Response: It is not clear how and why these findings are considered contradictory. In multiple places in the manuscript we indicate that once activated CFTR and NBCe1-B are NOT inhibited by depletion of PtdSer. We have an entire section showing the multiple experiments we did to show that PtdSer does not regulate the transporters conductance (pore properties), which led us to examine the role of PtdSer in signaling complexes formation and transduction.

7. Nanodomain Regulation by E-Syt3:

There is no evidence to support the claim that "the change in PtdSer is highly specific and localized in a subdomain accessed by E3C2C" (Page 6). With the current data, it remains unclear how E-Syt3 regulates PtdSer at the proposed nanodomains or whether such nanodomains even exist.

Response: We respectfully disagree with the assertion that we did not demonstrate the specificity of targeting by E3C2C. Our data show that only E3C2C and not E2C2C target the SMP domain at the junction to regulate the transporters. Furthermore, E3C2C uniquely targets all SMP domains of E-Syt1, E-Syt2 and E-Syt3 at the junction to similarly regulate the activity of the transporters.

Dear Shmuel,

We have now received re-review reports from two referees, which I have included below. As you will see, you have addressed their concerns satisfactorily; I am satisfied that the discussion over Referee #2 and #3's points on the specificity of the Evt probe will be available in the manuscript's accompanying online material. Before I can finally accept the manuscript, there are some remaining editorial points which need to be addressed. In this regard would you please:

- include up to five keywords,
- rename the conflict of interest statement, the "Disclosure and competing interests statement",
- remove the AC/CrediT section from the text,
- list all callouts sequentially; callout 8K should be corrected to Fig. 8K; in addition, there is a callout for a missing Table S1,
- specify section names in the 3rd (pink) column for positive response of the author checklist,
- save the Appendix file in PDF format; the title page should contain "Appendix for + ER/PM Junction PtdSer specified by E-Syt3/ORP5 regulate ion transporters and epithelial transport" and a table of contents with the page numbers for the listed items; nomenclature should be Appendix Figure Sx and Appendix Table Sx throughout the manuscript and Appendix PDF ,
- upload the Reagents and Tools table using the template from the guide to authors on our website,
- upload Source Data with all requested Source Data files specified in the blank Source Data checklist,
- specify p values in the legends of figures 1B iii, Diii, Iii, Jiii, 2 Biii, Cii, Diii, Eiii; 3A-E, Fiii, Giii, Hii; 4A iv, Ciii, D, E, Fii, I; 5D, F; 6B, 7A, C, F, I, J, K, L, O, P; 8G, 9B; EV1 B; EV3 Fii; EV4 G; EV8 A, EV9 B, EV10 A, EV11 C, D; EV12 A-C; EV13 E; EV14 C, D; EV 15A iii; EV16 A,
- specify the statistical test used for data analysis in the legends of figures 1 Biii, C, Diii, E, Hiii, Iii, Jiii, Kii; 2 Biii, Cii, Diii, Eiii; 3A-E, Fiii, Giii, Hii, Iii; 4A iv, Ciii, D, E, Fii, Hii, I, J; 5Aiii, Biii, Ciii, D, F, G; 6B, Ciii, Diii; 7A, B, C, F; Giii, Hiii, I, J, K, L, O, P; 8E-H, K-N; 9B, E; EV1 A, B, C, F, H, I; EV3 Fii, EV4 F, G; EV 6B; EV8 A, B; EV9 A, B; EV10 A-D; EV11 C, D; EV12 A-H; EV13 Aiii, Biii, Ci, iii, Dii, E; EV14 C, D; EV 15A iii, B iii; EV16 A,
- define 'n' in the legends of figures 1Biii, C, Diii, E, Hiii, Iii, Jiii, Kii; 2 Biii, Cii, Diii, Eiii; 3A-E, Fiii, Giii, Hii, Iii; 4A iv, Ciii, D, E, Fii, Hii, I, J; 5Aiii, Biii, Ciii, D, F, G; 6B, Ciii, Diii, I; 7A, B, C, F; Giii, Hiii, I, J, K, L, O, P; 8E-H, K-N; 9B, D, E; EV1 A, B, C, F, H, I; EV3 Fii; EV4 F-H; EV 6B, EV8 A, B; EV9 A, B; EV10 A-D; EV11 C-E; EV12 A-H; EV13 Aiii, Biii, Ci, iii, Dii, E; EV14 C, D; EV 15A iii, B iii; EV16 A; EV17 Aiii, Biii; EV18 B, C,
- define error bars in the legends of figures 1Biii, C, Diii, E, Hiii, Iii, Jiii, Kii; 2 Biii, Cii, Diii, Eiii; 3A-E, Fiii, Giii, Hii, Iii; 4A iv, Ciii, D, E, Fii, Hii, I, J; 5Aiii, Biii, Ciii, D, F, G; 6B, Ciii, Diii, I; 7A, B, C, F; Giii, Hiii, I, J, K, L, O, P; 8E-H, K-N; 9B, D, E; EV1 A, B, C, F, H, I; EV3 Fii; EV4 F-H; EV6 B, EV8 A, B; EV9 A, B; EV10 A-D; EV11 C-E; EV12 A-H; EV13 Aiii, Biii, E; EV14 C, D; EV 15A iii, B iii; EV16 A; EV17 Aiii, Biii; EV18 B, C, and
- correct the section order as follows: Title page - Abstract & Keywords - Introduction - Results - Discussion - Methods - Data Availability - Acknowledgements - Disclosure and Competing Interests Statement - References - Figure Legends - Table(s) - Expanded View Figure Legends.

We include a synopsis of the paper (see <http://emboj.embopress.org/>). Please provide me with a general summary image, a two sentence statement and 3-5 bullet points that capture the key findings of the paper.

I am looking forward to receiving your revised manuscript.

EMBO Press is an editorially independent publishing platform for the development of EMBO scientific publications.

Best wishes,

William

William Teale, PhD
Editor
The EMBO Journal
w.teale@embojournal.org

We realize that it is difficult to revise to a specific deadline. In the interest of protecting the conceptual advance provided by the work, we recommend a revision within 3 months (13th Jul 2025). Please discuss the revision progress ahead of this time with the editor if you require more time to complete the revisions. Use the link below to submit your revision:

Referee #1:

I have looked through the responses to my comments, as well as all the revisions to the Ms and I can confirm that the Ms has been significantly improved and that I have no further concerns. I would now recommend acceptance.

Referee #2:

The responses to my comments (Reviewer 2) are mostly satisfactory, though related to point 5, I still question whether the Evt probe is capable of sensing junctional PS as opposed to total PS pools on membranes. The authors' response did not address that issue. I would defer to Reviewer 3 to assess whether the responses to those substantial critiques were sufficient.

All editorial and formatting issues were resolved by the authors.

Dear Shmuel,

I am pleased to inform you that your manuscript has been accepted for publication in the EMBO Journal.

Congratulations to you and all of your coauthors!

Best wishes,

William

William Teale, PhD
Editor
The EMBO Journal
w.teale@embojournal.org
